# Bayesian Linear Regression Modelling for Sperm Quality Parameters Using Age, Body Weight, Testicular Morphometry, and Combined Biometric Indices in Donkeys

**DOI:** 10.3390/ani11010176

**Published:** 2021-01-13

**Authors:** Ana Martins-Bessa, Miguel Quaresma, Belén Leiva, Ana Calado, Francisco Javier Navas González

**Affiliations:** 1Department of Veterinary Sciences, School of Agrarian and Veterinary Sciences, University of Trás-os-Montes and Alto Douro, 5000-801 Vila Real, Portugal; miguelq@utad.pt (M.Q.); anacalad@utad.pt (A.C.); 2CECAV, Animal and Veterinary Research Center, University of Trás-os-Montes and Alto Douro, 5000-801 Vila Real, Portugal; 3Veterinary Teaching Hospital, University of Trás-os-Montes and Alto Douro, Quinta de Prados, 5000-801 Vila Real, Portugal; 4AEPGA-Association for the Study and Protection of Donkeys, Atenor, 5225-011 Miranda do Douro, Portugal; belenleiva.aepga@gmail.com; 5Genetics Department, Veterinary Sciences, Rabanales University Campus, University of Córdoba, Madrid-Cádiz Km. 396, 14014 Cordoba, Spain; fjng87@hotmail.com

**Keywords:** Bayesian models, testicular, sperm output, biometry, donkey

## Abstract

**Simple Summary:**

The prediction of sperm output and other reproductive traits based on testicular biometry is an important tool in the reproductive management of stallions. Nevertheless, corresponding research in donkeys remains scarce. Several donkey breeds in Europe face a compromising threat of extinction, which has been accelerated by the low renovation of populations and their inbreeding levels. Although research on female reproductive physiology has made crucial advances, much less is known about the physiology of the male. In the present work, two Bayesian models were built to predict for sperm output and quality parameters in donkeys. Models included combinations of age as a covariate and biometric and testicular measurements as independent factors. Results evidenced that the goodness-of-fit was similar for both models—hence, the combination of biometry and testicular factors presented improved predictive power. The application of these models may assist in the process of making decisions in respect to the reproductive/biological, clinical, and selection handling of the animals.

**Abstract:**

The aim of the present study is to define and compare the predictive power of two different Bayesian models for donkey sperm quality after the evaluation of linear and combined testicular biometry indices and their relationship with age and body weight (BW). Testicular morphometry was ultrasonographically obtained from 23 donkeys (six juveniles and 17 adults), while 40 ejaculates from eight mature donkeys were analyzed for sperm output and quality assessment. Bayesian linear regression analyses were considered to build two statistical models using gel-free volume, concentration, total sperm number, motility, total motile sperm, and morphology as dependent variables. Predictive model 1 comprised the covariate of age and the independent factors testicular measurements (length, height and width), while model 2 included the covariate of age and the factors of BW, testicular volume, and gonadosomatic ratio. Although goodness-of-fit was similar, the combination of predictors in model 1 evidenced higher likelihood to predict gel-free volume (mL), concentration (×10^6^/mL), and motility (%). Alternatively, the combination of predictors in model 2 evidenced higher predictive power for total sperm number (×10^9^), morphologically normal spermatozoa (%), and total motile sperm count (×10^9^). The application of the present models may be useful to gather relevant information that could be used hereafter for assisted reproductive technologies.

## 1. Introduction

Measuring testicular size reports an approximate measurement of the amount of testicular parenchyma present in a certain individual, which in turn determines the potential for sperm production [1,2]. Studies on equine testicular biometry are fairly common, and the correlation between testicular dimensions and the capacity for sperm production has frequently been addressed in the literature, allowing the establishment of a predictive formula for daily sperm output (DSO) [2,3,4]. Still, contextually, corresponding publications on male donkey remain scarce. Besides biometric testicular data such as length, width, and volume, other factors might be investigated as independent variables (covariates) in DSO prediction, like age, body weight (BW), or the gonadosomatic ratio (GSI), which is the gonadal weight/BW ratio [5]. Previous studies evidenced that knowledge from stallions could not be directly assumed for donkeys, as their reproductive physiology may present some specific particularities. For instance, donkeys present spermatogenesis cycles that last for 47.2 days, with each spermatogenic stage lasting for 10.5 days [6].

The numbers of the population of the Burro de Miranda’s (*Equus asinus*), as it occurs in other donkey breeds in southern Europe countries [7], dramatically fall, with a mature male population of around 40 jackasses and 300 breeding females. Besides, reproduction rates in such populations are low, with an excessive overuse of the same males in the past, which derived in the occurrence of genetic bottlenecks. Although the reproductive physiology of Miranda donkey females has already been studied [8], no research focusing on the biometry or reproductive physiology of Miranda donkey males has been reported to date. Considering the actual population structure, the interest in applying assisted reproductive technologies (ARTs) have raised. However, the implementation of these techniques requires consistent knowledge of reproductive biology in order to select the best males for ARTs programs to be successful. Testicular biometry, due to its correlation with sperm production, is a valuable tool to estimate male fertility and is also an essential element of the breeding soundness evaluation (BSE). In light of the aforementioned points, the accurate prediction of sperm output and quality parameters obtained from biometrical and testicular morphometric parameters, will improve the effectiveness of reproductive management in male donkeys.

In this context and bearing in mind the small but diverse mature male population, some issues in regards the statistical approach to follow may arise. For these reasons, the statistical tools used in this study were chosen to fit the characteristics of the data to be analyzed. According to Oravecz and Muth [9], the popularity of growth curve modelling (GCM) lies in its flexibility to simultaneously analyze within-individual changes (e.g., changes with age, change due to intervention, due to natural changes occurring along the life of the individuals, etc.) and between-individual effects (i.e., individual differences). In other words, GCM may be useful to model inter-individual differences and intra-individual variation. GCM has been successfully used to model the evolution of semen parameters in males from other species, such as boars [10].

An individual’s specific growth trajectory, specified as a mathematical function that describes how variables reciprocally relate over time, captures how an individual uniquely changes. GCM covers situations that range from those for which the change function is linear to other occasions when curvilinear polynomial functions are fitted (for instance, quadratic, cubic, etc.), which means that modelling is not limited to consider straight-line functional growth. Beyond handling varying growth functions, GCM can flexibly handle unbalanced designs, meaning study individuals may be measured at different occasions and need not be excluded from the analysis, even if some of their measurements are missing [8].

In these regards, Bayesian inference potentiates the flexibility of GCM, given Bayesian analyses do not assume large samples, as it would happen in maximum likelihood estimation (either it is nonparametric or parametric inference). Besides, smaller data sets can be evaluated preventing power loss and retaining precision, as suggested by Hox, et al. [11] and Lee and Song [12]. In small sample size conditions, the probability of finding significant results decreases [13]. Given power issues, this limitation often translates into an increased hardness to obtain meaningful results [14].

According to Stoltzfus [15], the basic assumptions that must be met for the outputs of regression analyses to be valid include independence of errors, linearity in continuous variables, the absence of multicollinearity, and a lack of strongly influential outliers. Additionally, there should be an adequate number of events per independent variable (covariate) to avoid an overfit model. Commonly recommended minimum “rules of thumb” range from 10 to 20 events per covariate. This would be supported by Chen, et al. [16], who suggested that the usual minimum number of observations for running a linear regression to be 30 to obtain statistically significant estimates. The same authors would even state that sometimes this requirement cannot be met, for instance when the number of individuals in the sample is limited, which is common to all donkey breeds [17]. Consequently, the general rule of thumb explains that, to succeed when conducting a linear regression analysis, the number of observations must not be smaller than 30 or 3× (k + 1), where k represents the number of independent variables (covariates); hence, the sample size used in the present study fulfils all the assumptions to be used in linear regression analyses.

Contextually, Bayesian estimation methods have been reported to require a much smaller ratio of parameters to observations (1:3 instead of 1:5); that is, Bayesian inference maximizes the ability to determine significant effects for relatively limited sample sizes. These sample limitations are reflected in the broadening of confidence intervals, which must be accompanied by an acceptable Bayes factor value.

To the best of our knowledge, no previous study has reported an estimation of donkey sperm output and quality traits using a Bayesian approach. In this context, the aims of the present study are to define and compare the predictive power of two Bayesian predictive models for sperm quality parameters using linear testicular measurements, combined biometrical indices, and their relationship with age and BW as predictive factors.

## 2. Materials and Methods

### 2.1. Animals

The study was carried out in the Veterinary Teaching Hospital of University of Trás-os-Montes and Alto Douro (VTH-UTAD, Vila Real, Portugal). Animals have been evaluated with approval and in collaboration with the Association for Study and Protection of the Donkey Breed Burro de Miranda (AEPGA), in the behalf of a scientific protocol of caooperation signed between both institutions. All animal procedures were conducted in accordance with national laws for animal welfare and experimentation as with the EU Directive 2010/63/EU for animal experiments and the approval of the Directive Hospital Committee (Approval Ref. 408/VTH-UTAD).

Animals were clinically examined, and the genital tract was palpated previous to ultrasonographic (US) evaluation. Body weight (BW) (kg) was assessed using an equine digital floor weighting scale. For the testicular morphometric evaluation, 23 Miranda donkeys were considered. Animals were allocated to two groups by age; juvenile to prepubertal (*n* = 6) (≤14 months) and mature (*n* = 17) (≥24 months) (Table 1). Only clinically healthy animals with normal size and consistency symmetrical testis and epididymis showing no echogenic changes in the testicular parenchyma were included. Epididymis and spermatic cords were included. Either in the juvenile or adult group, only animals with both testicles at scrotal position were considered. For the assessment of sperm, a sub-group of eight Miranda mature donkeys was selected from the mature group for further sperm collection and evaluation. Exams were performed during the autumn–winter season in 2018–2019 (US evaluations of the juveniles); and spring–summer in 2019 (semen collections and US examination of adult males).

### 2.2. Testicular Morphometry Evaluation

US testicular measurements were obtained from 23 donkeys aged from 7 to 259 months old and with 120 to 400 kg of weight (juvenile donkeys means: 11.17 ± 2.77 months old and 160.33 ± 24.66 kg; mature donkeys means: 79.94 ± 50.25 months old and 279.47 ± 54.43 kg, respectively). US measurements were performed with Philips^®^ CX30 Portable Ultrasound (Philips^®^, Amsterdam, Holland) with a sectorial 3.0–7.0 MHz transducer, following the previously described technique for stallion measurements [4]. Longitudinal and transversal plans were performed in each testicle, being the epididymis excluded from testicular US measurements. The electronic cursors were placed at the limit of tunica albuginea, and after three consecutive scans, the following parameters were obtained considering the largest measurement (cm): right and left length (L), height (H), and width (W) (cm). Right and left testicular volume (TV) were calculated using the Lambert formula, TV = L × W × H × 0.5233, used to measure the volume of an ellipsoid [4]. Total testicular volume (TTV), which represents the sum of the right and left TV, was obtained for each donkey. To compute gonadosomatic ratio (GSI) (%), i.e., testicular weight/BW, TTV (cm^3^) was directly converted into grams, based on the fact that testis volume density in mammals is very close to one [18]. After US measurements, routine orchiectomy was performed on five juvenile and two adult donkeys. After surgery, the extirpated testis (*n* = 14) were measured—the same measurements as in vivo—using precision sliding calipers.

### 2.3. Semen Collection and Evaluation

A sub-group of eight jackasses, ranging between 34 and 259 months of age (214–400 kg), was selected from the mature group for semen collection and further evaluation. A total of 40 ejaculates (five ejaculates per jackass) was collected. Donkeys had been successfully used in previous natural services. Before starting the experiment, sperm collections were performed for three consecutive days to minimize the number of sperm from extra-gonadal reserves, as it has been previously reported for donkeys [19]. Collections were performed at two-day intervals and using an artificial vagina (AV) (Hannover model—Minitub Iberica S.L., Tarragona, Spain) lubricated with non-spermicidal gel (ReproJelly—Minitub Iberica S.L., Tarragona, Spain), using a jenny in heat as a mount. The AV was filled with warm water to reach and maintain an inner temperature of 50–55 °C. A sterile semen collection bottle was used in each collection. The gel fraction was removed by filtering the whole ejaculate with a nylon filter (Minitub Iberica S.L., Tarragona, Spain). Gel-free ejaculate was immediately evaluated for volume (mL), motility (%), concentration (×10^6^/mL), and percentage of morphologically normal (%). Volume was measured in a graduated semen collection bottle. Then, each collected ejaculate was evaluated for sperm motility and concentration. For sperm motility evaluation, an aliquot of gel-free ejaculate was immediately extended 1:1 (vol/vol) with INRA 96 extender at 37 °C. Sperm motility was blind and subjectively estimated by the same experienced operator after the evaluation of motile spermatozoa (%) considering five different fields under light microscopy (×200), placing a semen droplet in a prewarmed (37 °C) slide covered by a cover slip. Concentration was determined using an improved Neubauer hemocytometer. Total sperm number (TSN, ×10^9^) was computed considering the product between the volume of gel-free ejaculates and sperm concentration, whereas total motile sperm count (TMS, ×10^9^) was obtained by computing the product between motility and TSN. Sperm morphology defects (head, intermediary piece, tail) were evaluated in eosin-nigrosin stained smears using light microscopy in oil immersion objective lens (×1000), counting a total of 200 sperm cells [20].

### 2.4. Statistical Analysis

#### 2.4.1. Parametric Assumptions Testing and Approach Decision

Since sample size was a limitation in this study, parametric assumptions were tested to decide on the most appropriate statistical approach to follow to analyse the present data. The Shapiro–Francia W’ test (for 50 < *n* < 2500 samples), Shapiro–Wilk test (for *n* < 50 samples), and Levene’s test were used to discard gross violations of parametric assumptions (normality and homoscedasticity). The Shapiro–Francia W’ test was performed using the Shapiro–Francia normality routine of the test and distribution graphics package of the Stata Version 15.0 software (StataCorp [21]. Appendix A report a gross violation of normality assumption occurred in all variables of testicular biometry and sperm parameters (*p* < 0.01), respectively, except for gel-free volume (mL) and sperm concentration (×10^6^/mL). Homoscedasticity was violated as well (*p* < 0.01); hence, a nonparametric approach was suggested.

All statistical tests, including all Bayesian procedures, were performed using the explore procedure of the descriptive statistics package in SPSS Statistics (Version 25.0, IBM Corp., Armonk, NY, USA) [22].

#### 2.4.2. Comparative Analysis of US and Caliper Testicular Morphometry between Juvenile and Mature Jacks

Bayesian one-way ANOVA procedure was used to detect differences in the means for testicular measurements between juvenile and mature jackstocks using the Bayesian ANOVA task from the Bayesian statistics procedure of SPSS Statistics, Version 25.0, IBM Corp. [22].

#### 2.4.3. Analysis of US Testicular Morphometry, Age and BW

Bayesian inference of Pearson’s correlation was used to characterize the posterior distribution of the linear correlation between age and BW, US testicular measurements, and composite indices using the Pearson correlation task from the Bayesian statistics procedure of SPSS Statistics, Version 25.0, IBM Corp. [22]. The correlation methods used and discussed in this paper can be validly used even if we work with repeated measures as we tested independent data [23]. Furthermore, in case variable pairs tested held a perfect linear correlation r_xy_ = 1, the integral equation to perform Bayesian inference for Pearson’s correlation would not have converged [24].

#### 2.4.4. Analysis of US and Caliper Testicular Morphometry

Bayesian inference of Pearson’s correlation was used to characterize the posterior distribution of the linear correlation between caliper testicular biometry variables (*n* = 14 testis) and US testicular biometry variables (*n* = 46 testis). The Pearson’s correlation coefficient measures the pairwise linear relation between the dependent variable y and the independent variable x. When r_xy_ = |1|, the dependent variable y is perfectly linearly correlated with the independent variable x. Then, following a decreasing order, a coefficient of |0.8| < r_xy_ <|1| suggests a strong linear correlation; a coefficient of |0.3| < r_xy_ <|0.6| suggests a moderate correlation; and a coefficient of 0 < r_xy_ < |0.3| suggests a weak correlation, respectivelyProfillidis and Botzoris [25].

The two methods were compared to decide on whether to use US or real biometric parameters or a combination of both to build Bayesian regression models. Bayesian inference for Pearson correlation was performed using the Pearson correlation task from the Bayesian statistics procedure of SPSS Statistics, Version 25.0, IBM Corp. [22]. The aforementioned test evidenced that US and caliper measuring methods were significantly correlated.

Appendix A summarizes the estimated Pearson’s correlation pairwise coefficients and respective Bayes factors. For all measurement pairs, the estimated Pearson’s correlation coefficient was always higher than 0.938, with corresponding Bayes factor of <0.001. As a result, the use of US measurements was exclusively selected to integrate the models for predicting sperm output, provided measurements were taken in vivo, hence they had a higher clinical applicability.

According to Doğan [26], although correlation analyses may erroneously detect the occurrence of incidental relationships instead of meaningful clinical/biological association, these may be a preferable choice under certain contexts. For instance, the same authors reported that one of the critical problems in other presumably more robust techniques such as the Bland–Altman analysis relies on the need for the data to meet the assumption of a normal distribution. Contrastingly, when testing Pearson’s correlations, the pairs of continuous variables need not be normally distributed, although their differences should. To determine the violation of this assumption, data may be tested against the normal distribution using classical methods such as the Shapiro–Wilk test or the Kolmogorov–Smirnov test.

Additionally, the same authors reported the fact that the Bland–Altman analysis is not an appropriate method to compare items for which repeated measurements were considered, as in the present study. In these regards, Batterham [27] would suggest that in a spreadsheet-based simulation of calibration and validity studies, a Bland–Altman plot of difference versus mean values for the instrument and criterion may show a systematic proportional bias in the instrument’s readings, even though none is present. This artifactual bias arises in a Bland–Altman plot of any measures with substantial random error. In contrast, a regression analysis of the criterion versus the instrument shows no bias. In this context, a regression analysis also provides complete statistics for recalibrating the instrument, if bias develops, or if random error changes since the last calibration. Consequently, the Bland–Altman analysis of validity should therefore be abandoned in favor of regression, as was performed in our study.

#### 2.4.5. Bayesian Linear Regression Modelling for Sperm Quality and Output Predictions

Gel-free volume (mL), concentration (×10^6^/mL), TSN (×10^9^), motility (%), morphologically normal (%), morphologically abnormal (%), gonadosomatic ratio (GSI) (%), and TMS (×10^9^) were considered the dependent variables in our study. Two separate statistical models were built, in which the predictive power of combinations of certain independent factors was evaluated.

Each of the regression models used in this study followed the general equation y_i_ = X_1_β_1_ + … X_i_β_i_ + ε_i_, where i = 1,2, … i is the ith number of factors; y_i_ is the vector of records for the aforementioned dependent variables with dimension n (217 records belonging to 31 jacks); X_i_ is the appropriate incidence matrix for factors; and β_i_ are the standardized regression coefficients for the ith number of factors and covariates considered, respectively. The general regression equation for model 1 was Y = Intercept + β_age (months)_·age (months) + β_Length LT (cm)_·length LT (cm) + β_Length RT (cm)_ ·length RT (cm) + β_Height LT (cm)·_ height LT (cm) + β_Height RT (cm)_· height RT (cm) + β_Width LT (cm)·_ width LT (cm) + β_Width RT (cm)_·width RT (cm). Oppositely, the general regression equation for model 2 was Y = Intercept + β_Age (months)_·Age (months) + β_BW (kg)_· BW (kg) + β_TTV (cm3)_·TTV (cm^3^) + β_GSI_·GSI, except for gonadosomatic ratio (GSI) (%), for which the last term in the equation was not included, provided this term refers to gonadosomatic ratio (GSI) (%) itself (β_GSI_·GSI).

According to Carlin [28], Bayesian inferences are sensitive to the dependence of variables on time (conditional on θ and x). If such dependence is large, it needs to be modeled, or the inferences will not be appropriate. For this reason, age was considered in the models. Under this design, time (age) plays a similar role to a blocking variable or covariable. For example, suppose that E(y|x, θ) has a linear trend in time (age) but that this dependence is not modeled (that is, suppose that a model is fit ignoring time (age)). Then, posterior means of factors or covariables in the model will tend to be reasonable, but posterior standard deviations will be too large, because this design yields treatment assignments that, compared to complete randomization, tend to be more balanced for time (age).

As Brewer [29] suggested, in our case, the use of an intercept was necessary as an empirical need for it was detected (for instance when unstandardized coefficients are used as in the present study). In these regards, confidence intervals for the estimated intercept were used as empirical indicators for the need of the intercept. Residual effects (ε_i_) were assumed to follow a normal distribution as $\varepsilon_{i}|X_{i}N\left(0,{\;\sigma }_{\varepsilon i}^{2}\right),$ where X_εi_ is an identity matrix and $\sigma_{\varepsilon i}^{2}$ is residual variance, respectively. For a continuous predictor variable, such as those in the present study, unstandardized coefficients are produced by the linear regression model using the independent variables measured in their original scales.

Unstandardized coefficients β_i_ can be interpreted considering what was stated by Hayes, et al. [30]—that all other variables being held constant, an increase of one unit in X_i_ is associated with an average increase of β_i_ units in Y. In the sections below, a detailed summary of the priors and posterior distributions used in this study is reported. A full description of the algorithms used by SPSS to perform Bayesian Inference on Multiple Linear Regression Models in this study can be found in the public document IBM SPSS Statistics Algorithms v. 25.0. by IBM Corp. [24].

When large number of parameters are being considered in a model, quadratic approximation has been reported to be computationally faster in terms of discretization and computing the likelihood over all possible parameter combinations compared to other approximations such as the Markov Chain Monte Carlo (MCMC) methods used in this study. However, the use of this quadratic approach was not feasible given it assumes the posterior distribution follows a normal distribution. In the context of our data, this assumption cannot be presumed provided the gross violation reported for the distribution properties reported at previous assumption testing stage.

After Bayesian Pearson’s correlation coefficients across variables had been performed, two distinct combinations of factors were evaluated. First, model 1 comprised the covariate of age (months) and the independent factors of LLT (length of left testicle) (cm), LRT (length of right testicle) (cm), HLT (length of left testicle) (cm), HRT (height of right testicle) (cm), WLT (cm) (width of left testicle), and WRT (width of right testicle) (cm). Second, model 2 comprised the covariate of age (months) and the factors BW (kg), TTV (total testicular volume) (cm^3^) and GSI (%). Lowest correlations were found for age and any of the rest of variables, hence, the covariate was retained in both models. The value of almost 1 for the correlation found between VLT (cm^3^) (volume of left testicle) and VRT (volume of right testicle) (cm^3^) and TTV (cm^3^) was the basis to decide on using composite TTV (cm^3^), given the reduced number of variables in model 2. BW was only considered in model 2, given the high generalized close to or above 0.9 correlations that it held with biometric caliper measurements. Bayesian linear regression analyses were performed using the linear regression package from the Bayesian statistics task of SPSS Statistics, Version 25.0, IBM Corp. [22]. The Bayesian Linear Regression test routine of the linear regression and related package of the Stata Version 16.0 software process was used to compute posterior distribution statistics for each factor included in each model to predict for each dependent variable. Once the analysis had been performed, we interpreted the estimated effect of the factors considered in the resulting predictive models, their confidence intervals, and the posterior distribution statistics.

#### 2.4.6. Jeffrey–Zellner–Siow (JZS) Mixture of g-Priors

For the present analyses, the Jeffrey–Zellner–Siow mixture of *g*-priors [31] was used. Jeffrey–Zellner–Siow’s prior somehow appears as a data-dependent prior through its dependence on X_i_, but this has been reported not to be a drawback since regression models are conditional on X_i_. As suggested by Heck [32], JZS prior could be an alternative that may satisfy several theoretical requirements such as the equality constraint on the test-relevant parameters, for instance of β, which leads to the null hypothesis H_0_ = β = β_0_ [33]. The benefits of JSZ prior distribution had also been reported by Rouder, et al. [34] and Liang, et al. [31]. Contextually, conditional on the residual variance ($\sigma_{\varepsilon i}^{2}$), the JZS prior defines a multivariate Cauchy distribution for the slope parameters of the full model, as follows

$(\beta_{i}|\sigma_{\varepsilon i}^{2})\sim \mathrm{MVC}\left(0_{P},\gamma_{i}{}^{2}\sigma_{\varepsilon i}^{2}C_{i}{}^{-1}\right)$, which is defined by a P-dimensional zero vector (location vector) and a scale matrix. The constant $\gamma_{i}$ determines the amount of scaling, which is chosen by the user a priori, the residual variance $\sigma_{\varepsilon i}^{2}$, and the matrix $C_{i}\;={\;X}_{i}^{\prime }X_{i}/N_{i}$, which is the covariance matrix of the centred design matrix $X_{i}$.

There are qualities of the JZS prior [34] that make it especially appropriate when performing linear regression analyses. Among these, the prior is symmetric and centered at zero in line with the predictive matching criterion as reported by Bayarri, et al. [35]. As a result, positive and negative values of the slope parameters have a priori the same probability to occur. Furthermore, JZS prior is scale invariant, thus the resulting Bayes factor does not depend on the scale of both the dependent variable and factors or covariates, hence results do not change when different unit variables are evaluated together, which is common in field conditions studies.

This independence from the measurements of model elements is achieved by scaling the multivariate Cauchy distribution by the residual variance $\sigma_{\varepsilon i}^{2}$ (a priori, a larger residual variance implies larger slopes) and by the inverse of the covariance matrix $C_{i}$ (a priori, a covariate with a larger variance implies smaller slopes). It may be worth considering that the procedure of defining a scaled prior for unstandardized coefficients (β_i_) equals the process of defining a prior for standardized coefficients $\left(\beta_{i}^{*}\right)$ [34].

Third, the scale parameter γ is fixed to a constant by the user, which allows prior beliefs to be specified about the expected effect size. IBM Corp. [24] algorithm guide reports that the algorithm of JZS prior for linear regression analyses, to compute the Bayes Factor uses the default value of $\gamma \;=\;2\sqrt{\pi }$ = 3.5, which reflects a prior belief of a medium effect size. For a single covariate x, this choice implies that the standardized regression slope $\beta_{i}^{*}\;={\;\beta }_{i}\cdot \;\mathrm{SD}\left(x_{i}\right)/\sigma_{i}$ has an a priori probability of 53.2% of being in the range [−0.50, +0.50].

Authors such as Rouder and Morey [36] also reported additional theoretical advantages of the JZS prior, such as its consistency in model selection (the fact that the Bayes factor, goes to infinity as the number of observations N increases without bound-favoring the data-generating model) or consistency in information (the Bayes factor for a certain effect goes to infinity as the proportion of explained variance or R Squared (R^2^) increases to 1). Additionally, Bayes factors for JZS prior can be relatively easily and highly precisely computed [37] and has been adapted for the default *t*-test [38], ANOVA [34], and linear regression [32].

#### 2.4.7. Factor and Covariate Effects Bayesian Modelling (FCEBM)

Being $y_{i}$, any of the effects of any of the independent variables (covariates) considered in this study, the posterior distribution of $y_{i}$ in the context of the data D is
$$p\left(y_{i}/D\right)\;=\;\sum_{i\;=\;20}^{i}p(y_{i}|M_{i},D)\;p(M_{i}|D)$$

This is an average of the posterior distributions of each model, weighted by the corresponding posterior model probabilities. In the previous equation, the posterior predictive distribution of $y_{i}$ given a particular model $M_{i}$ is
$$p\left(y_{i}|M_{i}D\right)\;=\;\int p\left(y_{i}|\beta_{i},M_{i},\;D\right)p(\beta_{i}|M_{i}D){d\beta }_{i}$$
and the posterior probability of model $M_{i}$ is given by
$$p\left(M_{i}|D\right)\;=\;\frac{p\left(D|M_{i}\right)p\left(M_{i}\right)}{\sum_{i\;=\;20}^{i}p\left(D|M_{i}\right)p\left(M_{i}\right)}$$
where
$$p\left(M_{i}|D\right)\;=\;\int p\left(D|\beta_{i},M_{i}\right)p(\beta_{i}|M_{i}){d\beta }_{i}$$
is the integrated likelihood of model $M_{i}$, $\beta_{i}$ is the vector of parameters of model $M_{i},p(\beta_{i}|M_{i})$ is prior density of $\beta_{i}$ under model $M_{i},p\left(D\left|\beta_{i}\right|M_{i}\right)$ is the likelihood, and $p\left(M_{i}\right)$ is the prior probability that $M_{i}$ is the true model.

For a problem with P potential covariates, the number of models, K, can be enormous (K = 2^P^ in the absence of other constraints). Only a small number of these models will have much support from the data, thus be selected by SPSS for each covariate. Marginal posterior distributions of all unknowns were estimated using the Gibbs sampling algorithm.

#### 2.4.8. Factors and Covariate Effect Bayesian Interpretation (CEBI)

The checklist proposed by Depaoli and Van de Schoot [39] was used to detect issues to check before estimating the model, (b) issues to check after estimating the model but before interpreting results, (c) understanding the influence of priors, and (d) actions to take after interpreting results.

Interpreting the effect of each particular covariate (independent variables used in this study) can be made as follows.

First, the posterior probability $p\left[\beta_{i}^{*}\neq 0/D\right]$ expresses the likelihood that the factor or covariate has an effect on each particular variable. Standard rules of thumb [40] for interpreting this posterior probability are as follows: <50% evidence against the effect; 50–75% weak evidence for the effect; 75–95% positive evidence; 95–99% strong evidence; >99% very strong evidence, whose results are comparable to commonly used thresholds to define significance of evidence through Bayes factor (BF) as reported in Appendix A.

Second, posterior distribution estimates (means) are used to measure the magnitude of the effect of a particular factor and covariate. For continuous predictor variables (metric covariates), such as the numeric variables used in this study, the regression coefficient represents the difference in the predicted value of the response variable for each one-unit change in the predictor variable, assuming all other predictor variables are held constant. When response variables are metric and can readily be interpreted in terms of impact, such as the ones in our study, β regression coefficients effect sizes by themselves.

Third, the 95% credibility interval shows that there is a 95% probability that these regression coefficients (posterior distribution mean value for each covariate and factor) in the population lie within the corresponding credibility intervals. When 0 is not contained in the credibility interval, a significant effect for such factor is detected.

Appendix A report a summary of posterior distribution statistics from Bayesian unstandardized linear (β) regression coefficients for each of the aforementioned variables considered in the analyses and a summary of Bayesian ANOVA outputs to test for differences in the means for US and caliper testicular measurements between juvenile (*n* = 6) and mature donkeys (*n* = 17).

#### 2.4.9. Convergence Criterion

The rounds of iteration continued until a tolerance convergence criterion of 10^−8^ was reached as suggested in literature [41]. Once the convergence criterion was reached, initial parameters were set, and model fitting properties were evaluated. The maximum number of iteration rounds used for each analysis was 2000 as suggested in IBM SPSS Statistics Algorithms version 25.0 by IBM Corp. [24]. This convergence criterion was chosen provided it has been used in Bayesian ANOVA and linear regression analyses in research contexts of limited sample sizes [42].

#### 2.4.10. Model Validity, Explanatory Power of Present Data, and Predictive Power of Future Data

The process of validation and comparison of Bayesian model is fully mathematically described in Geweke [43]. In this context, some authors [44] have suggested a correct proof for model validation should be based on the mean square error (MSE) of the models being evaluated. Additionally, although mean square residual or error (MSE) and minimum mean-square residual or error (MMSE) have been used and widely reported to measure how close a regression line is to a set of points (how good a certain model fits the data being observed), mean square prediction error or MSPE (= RSS/no. of observations) was chosen to measure error variation given MSE has been reported to be influenced by the number of predictors [26] in cases of reduced sample sizes [45,46].

Residual sum of squares (RSS) measures the amount of variance in a data set that is not explained by a regression model. That is, if we consider a regression to be a measure of the strength of the relationship between a dependent variable and an independent variable from a set of independent variables, then the RSS measures the amount of error remaining between the regression function and the data set—hence, it essentially determines how well a regression model explains or represents the data in the model. A smaller RSS figure represents a better suitability of the regression function to model for the data that it is intended to model.

In Bayesian inference, Monte Carlo Standard Error (MCSE) is another measure of accuracy of the chains. It is defined as the standard deviation of the chains divided by their effective sample size. MCSE has been reported to be the nonparametric or Bayesian counterpart of MSPE, and has been suggested to be used as the validation criteria in Bayesian Linear Regression model comparison studies [47].

Bayes factor (BF) provides an indirect measure of the explanatory power of the model to describe presently observed data (in our study). Larger BFs imply higher likelihoods for the combination of factors considered to explain the response variables being modelled. Commonly used thresholds to define significance of evidence following the premises by Jeffreys [48] and Lee and Wagenmakers [49] are reported in Appendix A. Intrinsically related to BF, Bayesian R^2^ can be considered as a data-based estimate of the proportion of variance explained for data. Additionally, acceptance rate, efficiency, and Monte Carlo standard error (MCSE) were used to determine the validity of the Bayesian methods implemented. Appendix A reports a summary of the description and interpretation of each model validity parameter used. Bayesian statistics predictive accuracy of the model [50] can be estimated through posterior predictive checking [51] (Appendix A).

BIC was then calculated, as it explains how well the model will predict on new data. Bayesian information criterion (BIC) or Schwarz information criterion (also SIC, SBC, SBIC) was computed as follows:(1)$$\mathrm{BIC}\;=\;N*N\;\ln\left(\mathrm{MSPE}\right)+K*\ln\left(N\right)$$
where MSPE is the mean squared prediction error, N is the number of observations or records, and K is the number of independent parameters of the model.

BIC was evaluated to compare predictive power across models. To summarize, BIC considers both the statistical goodness of fit and the number of parameters that have to be estimated to achieve this particular degree of fit, by imposing a penalty the number of parameters is increased [52,53]. BIC measures the trade-off between model fit and complexity of the model to determine [54]. Lower BIC values suggest that a particular model should have improved prediction properties in comparison to models for which higher values have been reported. In these regards, Bayesian R^2^ answers a different question as Bayesian R^2^ estimates the explanatory power of observed data, when the model is regression and non-adjusted R^2^ is used.

Frequently, when more variables are added, model predictive accuracy decreases. Consequently, a model with higher R^2^ will have higher-hence, worse-BIC values. The addition of “noise” variables to the fit (for which a relationship has not been suggested) will increase R^2^ values, but it will also decrease predictive power of the model. Hence, the model with more “noise” variables will have higher R^2^ and higher BIC.

## 3. Results

### 3.1. Descriptive Analysis for US Testicular Morphometry, Combined Biometric Indices and Sperm Output

Table 2 reports a summary of the descriptive statistics for age, BW, ultrasonographic (US) and caliper testicular morphometry (cm); L, H, W, TTV, GSI for juvenile and mature donkeys (*n* = 46 testis). Descriptive statistics of US and caliper measurements were computed for each group to perform a comparative analysis (Appendix A). In the juvenile group, mean TTV (cm^3^) was 17.74 ± 9.89 (*n* = 12 testis), while in mature group, TTV (cm^3^) was 271.69 ± 133.21 (*n* = 34 testis).

In the juvenile group, there was a progressive increase in all US testicular measurements, namely in TTV, from seven to 24 months, which was especially noticeable after 11 months of age. At 12–14 months, mean TTV was 21.05 ± 9.30 cm^3^ (*n* = 10 testis), and at 24–26 months, TTV was 85.27 ± 18.66 cm^3^ (*n* = 4 testis) (*P* < 0.01). Additionally, an increase in TTV was described after 150 kg of BW had been attained, which was verified in all donkeys after 12 months. On the other hand, after 168 months of age, a gradual decrease in TTV was noted. No difference between left and right testicle was found. Gonadosomatic ratio (GSI) (%) means in juveniles was 0.11 ± 0.06 and in matures 0.95 ± 0.39. Significant differences (*p* < 0.001) were found between juvenile and mature groups for all testicular biometrical parameters (Appendix A).

Results of sperm output and quality parameters (*n* = 40 ejaculates, observational unit); gel-free volume (mL), motility (%), concentration (×10^6^/mL), TSN (×10^9^), TMS (×10^9^), normal and abnormal sperm morphology (%) are presented in Table 1. TSN and TMS means was 18.453 ± 1.936 × 10^9^ sperm and 13.555 ± 1.479 × 10^9^ motile sperm, respectively. Sperm morphological abnormalities description can be consulted in Appendix A.

### 3.2. Statistical Analyses

#### 3.2.1. Bayesian Pearson’s Correlation Coefficients Preliminary Testing

Following a probabilistic view of regression, it can be assumed that any dependent variable (Y) has a certain associated variance σ². Linear regression bases on identifying the weight vector from observed data of a dependent variable to then use it to make predictions. For the model to be stable enough, the variance of the weight vector (W_ls_) should be low. If weight vectors variance is high, it means that the model is very sensitive to data. The weights differ largely with observed data if the variance is high. This means that the model might not perform well with observed data. When highly correlated covariables are used in regression models, the variance of the weight vector will be large. This occurs because when highly correlated features (covariates or factors) are considered, the values in the Singular Value Decomposition “S” matrix will be small. Hence inverse square of “S” matrix (S^−2^) will be large which makes the variance of W_ls_ large. For these reasons, Pearson’s correlation coefficients must be tested prior to performing regression analyses.

Table 3 summarizes the estimated sample Pearson’s correlation coefficient and the Bayes factors for BW (kg), age (months), US testicular biometric parameters and composite indices. For all variable pairs, the estimated Pearson’s correlation coefficient was always higher than 0.461, with a corresponding Bayes factor of <0.001, in all cases. Besides, moderate to high Bayesian inference Pearson’s correlation coefficients were found between age, BW, and testicular biometric variables. Pearson’s correlation coefficients between testicular biometry and BW were always >0.778, whereas Pearson’s correlation coefficients between testicular biometry and age were >0.467.

#### 3.2.2. Bayesian Linear Regression Modelling for Sperm Quality and Output Predictions

##### Model Explicative Power

Bayesian determination coefficients (R^2^) or percentages of variance captured for each of the two models and their respective Bayes factors are provided in Table 4. Both models were considerably more likely than others comprising just the intercept.

Bayesian estimates of linear regression coefficients for predictive models 1 and 2 for gel-free volume, concentration, morphologically normal or abnormal, TSN, GSI, motility, and TMS are presented in Table 5, Table 6 and Table 7. The intercept term in the regression evidences the average expected value for the response variable when all of the predictor variables are equal to zero.

##### Predictive Power and Model Validity

Posterior predictive *P* values for models 1 and 2 were around 0.331. The combination of predictors in model 1 evidenced a higher likelihood to predict for gel-free volume (mL), concentration (×10^6^/mL), and motility (%) (BIC: 387.587 to 534.480).

Yet, the combination of predictors in M=model 2 evidenced a higher likelihood to predict for TSN (×10^9^), morphologically normal and abnormal spermatozoa (%), TMS (×10^9^) and gonadosomatic ratio (GSI) (%), (BIC: −40.559 to 34,635.240). Age-related effects were verified on the following parameters: gel-free volume, morphologically abnormal spermatozoa (%), and TSN (Table 5 and Table 6). The summary of the results for the parameters of validity of both models is reported in Table 8.

## 4. Discussion

### 4.1. Testicular Morphometry (Juveniles and Matures) and Sperm Quality Parameters in Miranda Donkey Breed

The indication that testis biometry could provide a quantitative indication of sperm production has been previously reported in bulls [55,56], bucks [57,58], and dogs [59,60]. In the horse, morphometric, ultrasonographic-echotextural, and histomorphometric studies have been carried out [3,59,60], which evidenced the relation between testicular dimensions and sperm outputs [2] and the contribution of the ultrasonographic (US) evaluation in the accurate evaluation of the testicular functional status [61].

Albeit less than in horses, some studies on testicular morphometry have been conducted in donkey breeds such as Brazilian Pêga [6,62]; Ethiopian [63]; Egyptian [64,65]; and in the Italian breeds, Ragusano [19] and Martina Franca [66,67]. These studies have addressed the considerable existing variation among donkey breeds, which led to the need of investigating testicular dimensions in Miranda donkey breed. Besides, no previous works on Bayesian approaches to predict for sperm output and quality in donkeys has been conducted to the knowledge of the authors.

Mean US values of TTV of 271.69 cm^3^ (±133.21) obtained in mature Miranda donkeys were higher than those found in Egyptian donkeys [64], similar to those in Brazilian Pêga donkeys [62], and lower than those reported for Ethiopian donkeys [63]. In comparison to other morphologically similar breeds to Miranda donkey, our values were similar to slightly lower than TTV values found in Ragusano and Martina Franca donkeys [19,66]. Even if all the aforementioned breeds were medium to large-sized, differences of TTV could still be attributed to BW, age and management conditions of the males selected for the studies.

In the juveniles, studies are still scarce, but the values in our study (17.74 ± 9.89 cm^3^) were similar to those described for the prepubertal Egyptian [68] and Amiata donkeys [67]. Donkeys between 10 to 14 months are still in their pubertal transition period and, which reaches its end at 19–20 months of age, when testis have presumably completed their descent into the scrotum [37]. In the present work, a rapid increment of TV was verified after 11–12 months, which, besides, was simultaneous to the increase of BW. According to the work by Rota, et al. [67], a progressive increase of testicular width was noted after 10 months, and notably after 16 months of age; however, puberty—defined by the first presence in the ejaculate of 50 ×10^6^ sperm with at least 10% of motility-, was not attained in donkeys before 19–20 months. A previous histological work by our group evidenced that although a rapid increase of TV could be observed after 12–14 months, spermatogenesis was still incipient at that age [69]. Still, further studies should be carried out to precisely determine the age of Miranda donkey at puberty.

The comparative analysis of US measurements with those obtained using a precision caliper after orchiectomy evidenced that the former were very accurate. The precise position and orientation of the probe during US examination and the correct handling of the testicle, avoiding excessive tension during the exam, may have additionally contributed to the obtention of reliable US measurements.

The quantitative and qualitative sperm parameters obtained; total sperm number (TSN) per ejaculate, volume, concentration and morphology were within the range found for other European Donkey breeds such as Zamorano–Leonese [70], Catalonian [71], Andalusian [72,73,74], and Amiata donkey [67]. The values of GSI obtained for mature donkeys (0.9494) were higher than those reported in other domestic species [5]. This finding is of great interest and application when implementing ARTs’ strategies and is consistent with previous studies that observed the comparatively greater efficiency for sperm production of donkeys among mammals, characterizing by a high spermatogenic efficiency and short length of spermatogenesis [6,75].

### 4.2. Bayesian Approach and Predictive Models

Comparative observations were taken at different time points, from a population whose membership changes over time, but retains some constant members. This sample condition is known as partially overlapping samples and for this study, it implies the fact that not all the animals which were measured for morphometric parameters were evaluated for semen parameters. As reported by Kay, et al. [76] in studies working with partially overlapping samples, when there has been a gross violation of normality, as in our study, samples should be considered independent. In a nonparametric context, the strong subdivision of samples across the different experiments may condition results, hence, a Bayesian approach was followed given smaller data sets that can be evaluated avoiding power loss and retaining precision.

Posterior predictive *p* values (total model probability) for models 1 and 2 of around 0.331. Indicated moderately plausible good-fitting models. Similarly, the difference of more than 3 log likelihood units can be considered as strong evidence against models 1 or 2 depending on the parameters considered. The higher value reported for this parameter may suggest the acceptance of a more parameter-rich or simpler model accordingly. BIC explains how well the model will fit for new data (instead of explaining the existing data, which is measured by Adjusted R^2^). Models presenting lower BIC values evidence improved predictions for the dependent variable or variables that they model for. Frequently, adding more variables decreases predictive accuracy, and in that case, the model with even higher Adjusted R^2^ will display higher BIC, decreasing its predictive power [52,76,77]. However, considering the higher Adjusted R^2^ and the lower BIC, model 1 performs better at explaining and predicting than model 2 for gel-free volume (mL) and concentration (×10^6^/mL). For motility (%), model 1 was more precise to predict for future data while slightly worse at explaining present data (0.01 lower R^2^). The opposite situation was reported for TSN (×10^9^) and TMS (×10^9^), for which model 2 was more precise to predict for future data, although it may be slightly worse at explaining present data (0.13 lower R^2^). For morphologically normal and abnormal spermatozoa (%) and gonadosomatic ratio (GSI) (%), model 2 suggested a higher ability to explain for present and predict for future data.

In the present research, when comparatively analyzing testis’ biometry predictive power on spermatic parameters, some differences were found. The left testicle seems to exert a higher influence on gel-free volume, while, on the other hand, the biometry of the testicles seems to affect TSN differently. Concretely, the length and width of the left testicle and the height of the right testicle seem to increase in parallel with sperm quantitative parameters. Oppositely, as length and width of right testicle and height of the left testicle increase, sperms output parameters seem to decrease. Hence, the negative/positive balance between linear regression coefficients of morphometry variables (length, width and height) suggest that testis may reciprocally react to changes in the contralateral testicle, which affects almost all sperm outputs variables.

A previous work purposes the “compensation hypothesis” in birds, that states that one of the testis could serve as a “back-up” for any reduced function of the other and provides a mechanism to explain intraspecific variation in degree and direction of gonad asymmetry [77]. Another work relates that the degree of testicular asymmetry was positively correlated with inbreeding coefficient and negatively correlated with the proportion of normal sperm [78]. However, in the present work, testicular asymmetry was not found in both clinical and morphometric evaluation, as both features do not meet the inclusion criterion.

Mahmoud Ali Omar, et al. [79] reported a similar compensatory effect in the right testicle after the removal of the contralateral testicle in donkeys. Other authors have ascribed this compensation to the increase in serum LH and FSH concentrations and, potentially higher intratesticular testosterone [80]. Unilateral orchiectomy has been reported to increase the mean diameter of seminiferous tubules by 21% and of their lumina by 51% [81]. Additionally, two events in line with our results were described. A weight compensation was reported for the remaining testis, which has been already described [82]. Also, the histological examination of the testis of donkeys after unilateral orchiectomy with scrotum suture revealed hyperplasia of Leydig and Sertoli cells [79]. This had also been reported by Putra and Blackshaw [83], who suggested an increase in the number of Sertoli cells and germ cells occupying the seminiferous epithelium after unilateral orchiectomy. Our results may evidence that compensation may occur physiologically without these events, as it has also been reported in other species [78]. Still, future works are necessary in order to confirm these findings in donkeys.

In the present study, the age covariate, included in both predictive models, was significantly and positively correlated with several parameters, namely with gel-free volume and sperm output (TSN). The significant age-related positive effects on gel-free volume and TSN agreed those in previous works in stallions [84,85]. For instance, the influence of age in testicular dimensions of juvenile and peripubertal donkeys was verified by Rota, et al. [67], who suggested that age markedly influenced testicular width.

On the other hand, age shows a linear association with abnormal sperm morphology in model 1. Morphologically abnormal spermatozoa percentage slightly increases with age; while sperm concentration and morphologically normal spermatozoa linearly decrease. The negative impact of advanced age on morphology has been already described in stallions and has been ascribed to testicular degeneration, abnormal epididymal function [86] or to age-related testicular dysfunction associated with deterioration in DNA sperm motility [87]. A study in Egyptian donkeys reports that from six years onward, histological features were indicative of spermatogenic efficiency starting to decrease [65]; however, more studies should be performed before concluding that the same occurs in Miranda donkey breed.

In general, stronger correlations between BW and testicular biometry than between age and testicular biometry were verified in the present study. This agrees with the findings in a previous study conducted in stallions which emphasized the influence of body size in testicular measurements and sperm output [2]. However, the analysis of regression coefficients evidenced that the association of motility and total motile sperm (TMS) with TTV was not always constant. On the contrary, sperm motility, as well as TMS and concentration, were positively and linearly associated with gonodasomatic ratio (GSI). Overall, this supports the fact that even if the measurements of the testicular parameters could provide useful information about the potential sperm production, when it comes to predict motility, these parameters should be adjusted for the BW of the donkey, as reported by [Woodall and Johnstone [88]] when predicting for fertility in dogs. Contextually, further investigations should allow to determine and confirm the relationship between BW and TV in donkeys.

## 5. Conclusions

The results of the present work evidence the reliability of ultrasonographic measurements of testis, which emphasizes its importance and value to obtain reference values of donkey testicular volumes. Values of testicular volume and sperm output in the Miranda donkey breed are similar to those in other affine European donkey breeds. Gonadomatic index (GSI) is higher in the donkey than in other domestic species as previously described, which confirms the great reproductive potential of male donkeys.

Combinations of biometrical and testicular morphometric factors (age, body weight, testicular volume and GSI) will likely improve the predictive accuracy of Models than using factors separately. Besides biometry, considering data such as BW and age, testicular volume, and GSI may be systematically taken into consideration and integrated on BSE of donkeys. The present study provides new insights into donkey reproductive biology, which may be transferred to ARS strategies. Appropriate use of both models may be useful to further improve knowledge on the reproductive characteristics of donkey breeds, which may reinforce clinical purposes and maximize the outcomes from direct conservation or selection strategies.

## Figures and Tables

**Table 1 animals-11-00176-t001:** Donkeys enrolled in the study.

N	Age Range	Age Percentile	Median Weight Evolution Per Percentile
6	7 to 14 months	14 months (P25)	200 kg
11	15 to 95 months	40 months (P50/Median)	248 kg
6	≥96 months	96 months (P75)	302 kg

P25: value at 25% of observations; P50: value at 50% of observations and P75: value at 75% of observations.

**Table 2 animals-11-00176-t002:** Descriptive statistics for testicular US measurements in 23 donkeys (*n* = 46 testis, observational sample) and precision caliper after orchiectomy in seven of these donkeys (*n* = 14 testis, observational sample) and sperm quality parameters (*n* = 40 ejaculates, observational sample).

Items	N	Mean	SEM	SD	Skewness	Kurtosis	Minimum	Percentile 25	Median	Percentile 75	Maximum
Body Weight (kg)	161	248.39	5.63	71.38	0.23	−0.56	120.00	200.00	248.00	302.00	400.00
Age (months)	161	62.00	4.66	59.07	1.77	3.28	7.00	14.00	40.00	96.00	259.00
US Length LT (cm)	161	6.94	0.19	2.42	−0.51	−1.07	2.80	3.87	7.50	8.76	10.60
US Length RT (cm)	161	6.81	0.20	2.51	−0.50	−1.07	2.36	4.10	7.57	8.87	10.10
US Height LT (cm)	161	4.16	0.12	1.57	−0.11	−1.07	1.50	2.63	4.51	5.50	6.93
US Height RT (cm)	161	3.96	0.12	1.48	0.15	−0.01	1.40	2.56	4.26	4.94	7.61
US Width LT (cm)	161	5.18	0.16	1.98	−0.44	−1.11	1.50	3.32	5.42	6.86	7.88
US Width RT (cm)	161	5.17	0.16	1.97	−0.37	−1.13	1.60	3.07	5.69	6.69	8.40
US Volume LT (cm^3^)	161	106.60	6.56	83.22	0.45	−0.82	3.30	17.95	95.58	175.20	283.91
US Volume RT (cm^3^)	161	98.88	6.13	77.80	0.65	−0.10	2.93	19.28	91.39	136.63	297.64
US TTV (cm^3^)	161	205.44	12.62	160.08	0.52	−0.53	6.23	37.23	185.30	329.08	581.54
US GSI (%)	161	0.73	0.04	0.50	0.37	−0.60	0.04	0.22	0.72	1.09	1.86
Caliper Length LT (cm)	49	4.70	0.27	1.86	1.24	−0.06	3.30	3.50	3.70	6.40	8.50
Caliper Length RT (cm)	49	4.83	0.29	2.06	1.13	−0.13	3.00	3.20	4.00	6.60	9.00
Caliper Height LT (cm)	49	3.13	0.13	0.94	0.61	−1.37	2.20	2.30	2.70	4.50	4.50
Caliper Height RT (cm)	49	3.06	0.15	1.05	0.51	−1.46	2.00	2.00	2.50	4.50	4.60
Caliper Width LT (cm)	49	3.23	0.20	1.41	1.00	−0.37	1.90	2.00	2.50	4.50	6.00
Caliper Width RT (cm)	49	3.26	0.23	1.63	1.09	−0.31	1.90	2.00	2.50	4.80	6.50
Caliper Volume LT (cm^3^)	49	35.55	5.73	40.10	1.34	0.25	7.94	8.35	14.13	67.81	120.10
Caliper Volume RT (cm^3^)	49	38.85	6.70	46.93	1.33	0.25	6.59	7.33	10.46	76.25	137.76
Caliper TTV (cm^3^)	49	74.40	12.43	86.99	1.34	0.26	14.53	17.89	21.46	144.06	257.86
Caliper GSI (%)	49	0.35	0.05	0.35	1.16	−0.25	0.10	0.10	0.14	0.69	1.05
Gel-free volume (mL)	40	75.09	6.49	41.07	0.51	0.19	12.00	38.25	75.25	103.50	189.00
Concentration (× 10^6^/mL)	40	281.00	21.03	133.00	0.02	−0.41	45.00	213.75	282.50	363.75	540.00
TSN (× 10^9^) sperm	40	18.45	1936.98	12,250.54	1.27	2.05	4560.00	8482.50	15,750.00	25,653.75	59,360.00
Motility (%)	40	72.13	2.60	16.44	−1.35	1.80	20.00	60.00	77.50	85.00	90.00
Morphologically normal sperm (%)	40	87.35	1.58	9.97	−1.46	1.95	58.00	83.00	90.00	94.00	99.00
Morphologically abnormal sperm (%)	40	12.43	1.52	9.61	1.52	2.45	1.00	6.00	10.00	17.00	42.00
TMS (× 10^9^) sperm	40	13,555.38	1479.60	9357.81	0.91	0.76	1650.00	5607.75	11,264.50	20,300.00	42,642.00

LT—left testicle; RT—right testicle; TTV—total testicular volume; BW—body weight; TSN—total sperm number; GSI—gonadosomatic ratio; TMS—total motile sperm count.

**Table 3 animals-11-00176-t003:** Bayesian inference Pearson’s correlation output summary for BW (kg), age (months), US parameters, and composite indices.

	Body Weight (kg)	Age (months)	Length LT (cm)	Length RT (cm)	Height LT (cm)	Height RT (cm)	Width LT (cm)	Width RT (cm)	Volume LT (cm^3^)	Volume RT (cm^3^)	TTV (cm^3^)	GSI (%)
Body Weight (kg)	1.000	0.552	0.845	0.876	0.825	0.778	0.826	0.824	0.797	0.805	0.806	0.680
Age (months)	0.552	1.000	0.511	0.600	0.479	0.523	0.522	0.556	0.467	0.539	0.505	0.461
Length LT (cm)	0.845	0.511	1.000	0.977	0.944	0.924	0.942	0.925	0.916	0.908	0.918	0.917
Length RT (cm)	0.876	0.600	0.977	1.000	0.948	0.897	0.957	0.946	0.917	0.911	0.920	0.903
Height LT (cm)	0.825	0.479	0.944	0.948	1.000	0.908	0.940	0.926	0.962	0.922	0.949	0.939
Height RT (cm)	0.778	0.523	0.924	0.897	0.908	1.000	0.901	0.858	0.897	0.930	0.919	0.914
Width LT (cm)	0.826	0.522	0.942	0.957	0.940	0.901	1.000	0.962	0.922	0.901	0.917	0.923
Width RT (cm)	0.824	0.556	0.925	0.946	0.926	0.858	0.962	1.000	0.903	0.900	0.908	0.909
Volume LT (cm^3^)	0.797	0.467	0.916	0.917	0.962	0.897	0.922	0.903	1.000	0.976	0.994	0.964
Volume RT (cm^3^)	0.805	0.539	0.908	0.911	0.922	0.930	0.901	0.900	0.976	1.000	0.993	0.951
TTV (cm^3^)	0.806	0.505	0.918	0.920	0.949	0.919	0.917	0.908	0.994	0.993	1.000	0.964
GSI (%)	0.680	0.461	0.917	0.903	0.939	0.914	0.923	0.909	0.964	0.951	0.964	1.000

BF < 0.0001; GSI—gonadosomatic ratio (%).

**Table 4 animals-11-00176-t004:** Bayes Factor Model Summary for model 1 (comprising age and testicular morphometric parameters) and model 2 (comprising age, BW, TTV, and GSI) to predict for sperm output and quality in Miranda donkey breed.

**Model 1**	**Bayes Factor**	**R**	**R Squared**	**Adjusted R Squared**
Gel-free volume (mL)	406,756.54	0.855	0.731	0.682
Concentration (×10^6^/mL)	1554.89	0.788	0.621	0.553
TSN (×10^9^)	1308.11	0.786	0.617	0.548
Motility (%)	180.53	0.754	0.568	0.490
Morphologically normal (%)	47,305.85	0.832	0.693	0.637
Morphologically abnormal (%)	8839.07	0.812	0.660	0.598
GSI	1.38 × 10^19^	0.980	0.961	0.954
TMS (×10^9^)	52,401.57	0.833	0.695	0.639
**Model 2**	**Bayes Factor**	**R**	**R Squared**	**Adjusted R Squared**
Gel-free volume (mL)	252,538.00	0.794	0.630	0.599
Concentration (×10^6^/mL)	1169.55	0.706	0.498	0.457
TSN (×10^9^)	370.37	0.682	0.465	0.420
Motility (%)	5160.20	0.734	0.539	0.500
Morphologically normal (%)	1,907,536.17	0.819	0.670	0.643
Morphologically abnormal (%)	259,329.18	0.794	0.631	0.600
GSI	4.89 × 10^44^	0.999	0.998	0.998
TMS (×10^9^)	7080.51	0.740	0.547	0.509

TSN—total sperm number; TMS—total motile sperm count; GSI—gonadosomatic ratio.

**Table 5 animals-11-00176-t005:** Bayesian Estimates of Unstandardized Linear Regression Coefficients for predictive model 1 for gel-free volume (mL), concentration (×10^6^/mL), morphologically normal (%), and morphologically abnormal (%) sperm output in Miranda donkey breed.

Parameter	Posterior			95% Credible Interval		Parameter	Posterior			95% Credible Interval	
Gel-free volume (mL)	Mean	SD	MCSE	Lower Bound	Upper Bound	Morphologically normal (%)	Mean	SD	MCSE	Lower Bound	Upper Bound
(Intercept)	6.750	85.650	7.882	11.940	−159.039	(Intercept)	38.820	61.999	17.656	35.535	−69.634
Age (months)	0.554	0.186	0.020	0.551	0.190	Age (months)	0.009	0.120	0.033	0.016	−0.273
Length LT (cm)	−7.618	23.395	2.210	−8.748	−52.776	Length LT (cm)	7.176	15.183	4.290	8.154	−29.472
Length RT (cm)	−8.047	19.952	1.350	−7.762	−45.424	Length RT (cm)	−7.758	6.349	0.756	−7.367	−21.025
Height LT (cm)	39.055	10.577	0.513	39.257	16.789	Height LT (cm)	4.253	2.752	0.142	4.226	−1.076
Height RT (cm)	−0.142	15.488	1.598	−0.446	−30.291	Height RT (cm)	−2.344	10.261	2.868	−3.001	−20.264
Width LT (cm)	−8.293	13.214	1.203	−8.355	−35.057	Width LT (cm)	2.925	5.712	1.243	3.167	−7.979
Width RT (cm)	−0.867	16.024	1.877	−0.780	−32.031	Width RT (cm)	3.021	6.681	1.586	2.709	−10.081
Concentration (× 10^6^/mL)	Mean	SD	MCSE	Lower Bound	Upper Bound	Morphologically abnormal (%)	Mean	SD	MCSE	Lower Bound	Upper Bound
(Intercept)	119.325	95.837	6.792	120.692	−67.802	(Intercept)	−92.566	4.923	1.290	−92.168	−102.543
Age (months)	−1.675	0.438	0.024	−1.672	−2.510	Age (months)	0.268	0.036	0.002	0.267	0.199
Length LT (cm)	−179.851	51.735	14.252	−170.219	−289.282	Length LT (cm)	28.736	4.098	0.370	28.543	21.311
Length RT (cm)	112.455	63.065	7.805	114.114	−13.492	Length RT (cm)	1.078	5.696	0.306	1.182	−10.506
Height LT (cm)	−82.447	38.657	3.239	−84.282	−154.495	Height LT (cm)	−5.022	2.742	0.145	−5.009	−10.399
Height RT (cm)	60.562	33.694	8.149	57.459	2.457	Height RT (cm)	−22.037	2.535	0.272	−21.978	−27.089
Width LT (cm)	143.483	47.524	9.710	138.746	58.199	Width LT (cm)	8.980	3.366	0.145	8.888	2.389
Width RT (cm)	12.984	51.340	8.872	9.472	−80.658	Width RT (cm)	−15.696	3.963	0.276	−15.608	−23.572

LT—left testicle; RT—right testicle.

**Table 6 animals-11-00176-t006:** Bayesian Estimates of Unstandardized Linear Regression Coefficients for predictive model 1 for TSN, gonadosomatic ratio (GSI) (%), motility (%), and TMS (×10^9^) sperm output in Miranda donkey breed.

Parameter		Posterior		95% Credible Interval		Parameter		Posterior		95% Credible Interval	
TSN	Mean	SD	MCSE	Lower Bound	Upper Bound	Gonadosomatic ratio (GSI)	Mean	SD	MCSE	Lower Bound	Upper Bound
(Intercept)	23,753.340	80.646	4.744	23,753.840	23,598.750	(Intercept)	−0.600	0.035	0.003	−0.602	−0.670
Age (months)	−1416.025	443.100	133.800	−1451.571	−2085.682	Age (months)	0.000	0.000	0.000	0.000	−0.001
Length LT (cm)	8883.943	111.914	30.604	8885.097	8667.620	Length LT (cm)	0.048	0.031	0.003	0.046	−0.014
Length RT (cm)	13,450.930	78.239	21.894	13,444.170	13,319.860	Length RT (cm)	−0.086	0.036	0.004	−0.086	−0.156
Height LT (cm)	−33,418.180	131.843	37.637	−33,411.060	−33,681.770	Height LT (cm)	0.154	0.029	0.003	0.153	0.099
Height RT (cm)	3060.495	62.605	4.230	3059.644	2940.727	Height RT (cm)	0.097	0.027	0.002	0.097	0.046
Width LT (cm)	19,157.550	69.930	5.174	19,157.660	19,020.890	Width LT (cm)	0.036	0.030	0.003	0.037	−0.034
Width RT (cm)	−17,716.000	24.807	1.749	−17,716.020	−17,763.690	Width RT (cm)	0.072	0.025	0.002	0.071	0.026
Motility (%)	Mean	SD	MCSE	Lower Bound	Upper Bound	TMS (× 10^9^)	Mean	SD	MCSE	Lower Bound	Upper Bound
(Intercept)	−12.707	90.778	13.158	−11.443	−185.472	(Intercept)	−782.074	1094.008	328.918	−1260.849	−1857.071
Age (months)	−0.069	0.175	0.022	−0.070	−0.390	Age (months)	−612.420	250.955	74.637	−643.819	−958.823
Length LT (cm)	16.644	21.823	3.203	15.668	−24.687	Length LT (cm)	−3816.467	87.995	16.210	−3819.298	−3982.185
Length RT (cm)	12.235	11.157	0.668	11.796	−9.577	Length RT (cm)	4859.292	206.757	60.216	4806.093	4601.664
Height LT (cm)	−5.247	5.292	0.369	−5.314	−15.183	Height LT (cm)	−2809.112	2318.070	702.958	−1870.947	−7896.015
Height RT (cm)	−11.939	14.762	2.118	−11.310	−40.592	Height RT (cm)	536.574	140.014	15.828	524.424	295.207
Width LT (cm)	2.605	9.550	0.603	2.700	−16.521	Width LT (cm)	1678.912	2687.695	816.107	484.096	−742.483
Width RT (cm)	−13.494	10.278	0.916	−13.619	−33.146	Width RT (cm)	−7924.881	837.695	252.844	−7632.309	−9824.660

**Table 7 animals-11-00176-t007:** Bayesian Estimates of Unstandardized Linear Regression Coefficients for predictive model 2 for gel-free volume (mL), concentration (×10^6^/mL), morphologically normal (%) and morphologically abnormal (%), TSN, gonadosomatic ratio (GSI) (%), motility (%), and TMS (×10^9^) sperm output in Miranda donkey breed.

Parameter		Posterior		95% Credible Interval		Parameter		Posterior		95% Credible Interval	
Gel-free volume (mL)	Mean	SD	MCSE	Lower Bound	Upper Bound	Morphologically normal (%)	Mean	SD	MCSE	Lower Bound	Upper Bound
(Intercept)	18.027	74.948	4.027	17.645	−131.021	(Intercept)	101.165	48.993	2.230	102.862	6.593
Age (months)	0.380	0.075	0.003	0.386	0.228	Age (months)	−0.088	0.028	0.001	−0.089	−0.141
BW (kg)	0.059	0.241	0.013	0.057	−0.408	BW (kg)	−0.048	0.153	0.007	−0.051	−0.353
TTV (cm^3^)	0.229	0.206	0.011	0.237	−0.183	TTV (cm^3^)	0.082	0.130	0.006	0.086	−0.178
GSI	−61.537	63.042	3.287	−61.751	−178.796	GSI	−16.479	39.912	1.736	−17.993	−95.823
Concentration (× 10^6^/mL)	Mean	SD	MCSE	Lower Bound	Upper Bound	Morphologically abnormal (%)	Mean	SD	MCSE	Lower Bound	Upper Bound
(Intercept)	34.599	87.218	4.216	29.855	−135.220	(Intercept)	−34.631	48.161	2.471	−33.664	−130.104
Age (months)	−1.281	0.246	0.012	−1.287	−1.751	Age (months)	0.098	0.027	0.002	0.098	0.045
BW (kg)	1.577	0.397	0.018	1.568	0.802	BW (kg)	0.151	0.151	0.008	0.147	−0.133
TTV (cm^3^)	−0.434	0.282	0.013	−0.443	−1.002	TTV (cm^3^)	−0.167	0.128	0.006	−0.168	−0.424
GSI	71.418	77.349	4.329	72.597	−84.034	GSI	43.228	39.364	1.902	43.112	−30.970
TSN	Mean	SD	MCSE	Lower Bound	Upper Bound	Gonadosomatic ratio (GSI) (%)	Mean	SD	MCSE	Lower Bound	Upper Bound
(Intercept)	11,264.220	1978.212	598.275	10,786.800	8794.520	(Intercept)	1.213	0.036	0.001	1.214	1.140
Age (months)	4577.180	417.008	125.933	4671.496	3705.060	Age (months)	−0.001	0.000	0.000	−0.001	−0.001
BW (kg)	−5091.494	439.681	133.092	−5203.422	−5634.810	BW (kg)	−0.004	0.000	0.000	−0.004	−0.004
TTV (cm^3^)	2606.332	220.725	66.742	2662.469	2131.406	TTV (cm^3^)	0.003	0.000	0.000	0.003	0.003
GSI	−4379.107	756.935	227.540	−4209.127	−5956.150						
Motility (%)	Mean	SD	MCSE	Lower Bound	Upper Bound	TMS (× 10^9^)	Mean	SD	MCSE	Lower Bound	Upper Bound
(Intercept)	10.199	65.301	3.019	8.928	−114.356	(Intercept)	−4076.090	195.502	54.547	−4028.317	−4586.811
Age (months)	−0.116	0.043	0.002	−0.116	−0.202	Age (months)	−1422.795	311.451	88.829	−1524.509	−1776.786
BW (kg)	0.223	0.206	0.010	0.221	−0.173	BW (kg)	569.508	82.412	14.404	572.642	397.312
TTV (cm^3^)	−0.152	0.175	0.008	−0.156	−0.484	TTV (cm^3^)	−172.885	46.034	7.844	−172.536	−261.502
GSI	51.775	53.600	2.428	54.210	−54.645	GSI	2298.200	133.412	29.547	2281.863	2070.603

TTV—total testicular volume; TSN—total sperm number; TMS—total motile sperm count; GSI—gonadosomatic ratio.

**Table 8 animals-11-00176-t008:** Model validity parameters.

**Model 1**	**Gel-Free Volume (mL)**	**Concentration (× 10^6^/mL)**	**TSN**	**Motility (%)**	**Morphologically Normal (%)**	**Morphologically Abnormal (%)**	**Gonadosomatic Ratio (GSI)**	**TMS (× 10^9^)**
MCMC iterations	12.500	12.500	12.500	12.500	12.500	12.500	12.500	12.500
Burn-in	2.500	2.500	2.500	2.500	2.500	2.500	2.500	2.500
MCMC sample size	10.000	10.000	10.000	10.000	10.000	10.000	10.000	10.000
Number of obs	40.000	40.000	40.000	40.000	40.000	40.000	40.000	40.000
Acceptance rate	0.299	0.225	0.800	0.314	0.325	0.292	0.345	0.539
Min efficiency	0.007	0.001	0.001	0.005	0.001	0.001	0.015	0.001
Avg efficiency	0.023	0.010	0.011	0.019	0.008	0.029	0.037	0.002
Max efficiency	0.083	0.033	0.029	0.065	0.037	0.059	0.071	0.008
Log marginal likelihood	−209.558	−259.862	−132,346.360	−186.416	−164.579	−166.624	−31.747	−5993.399
BIC	433.872	534.480	264,707.476	387.587	343.913	348.003	78.249	12,001.553
**Model 2**	**Gel−Free Volume (mL)**	**Concentration (× 10^6^/mL)**	**TSN**	**Motility (%)**	**Morphologically Normal (%)**	**Morphologically Abnormal (%)**	**Gonadosomatic Ratio (GSI)**	**TMS (× 10^9^)**
MCMC iterations	12.500	12.500	12.500	12.500	12.500	12.500	12.500	12.500
Burn-in	2.500	2.500	2.500	2.500	2.500	2.500	2.500	2.500
MCMC sample size	10.000	10.000	10.000	10.000	10.000	10.000	10.000	10.000
Number of obs	40.000	40.000	40.000	40.000	40.000	40.000	40.000	40.000
Acceptance rate	0.302	0.410	0.467	0.348	0.353	0.402	0.300	0.705
Min efficiency	0.032	0.032	0.001	0.036	0.045	0.031	0.049	0.001
Avg efficiency	0.049	0.056	0.001	0.051	0.063	0.049	0.064	0.002
Max efficiency	0.108	0.125	0.001	0.088	0.129	0.099	0.095	0.003
Log marginal likelihood	−215.503	−263.929	−17,310.242	−186.604	−163.466	−163.531	27.657	−1746.014
BIC	445.762	542.613	34,635.240	387.964	341.687	341.817	−40.559	3506.783

## Data Availability

Data available on request due to restrictions eg privacy or ethical.

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
