# Peer review of "Bayesian Linear Regression Modelling for Sperm Quality Parameters Using Age, Body Weight, Testicular Morphometry, and Combined Biometric Indices in Donkeys"

_animals, 2021, doi:10.3390/ani11010176_

Round 1
Reviewer 1 Report
The authors do not accept that it is necessary to remove data from donkeys where testicular descent was not completed. They argue that testicular descent in donkeys does usually not occur before puberty. This is not acceptable: in donkeys, the percentage of males with cryptorchism is much higher than in horses (e.g. >30% in a study including 274 donkeys in Africa, Kumi-Diaka et al., Theriogenology 1981 VOL. 15 NO. 3, 241-243). Testicular descent in equids is not related to puberty and e.g. in horses occurs at birth whereas puberty occurs at an age of approximately 12 to 18 months. In male donkeys, puberty may also occur at a much earlier age (e.g. Sprayson and Thiemann, In Practice 2007; doi 10.1136/inpract.29.9.526). The authors of the present manuscript can therefore certainly not predict that testicular descent will occur in all prepubertal animals included into the study. These data therefore have to be removed from the analysis.
Changes and comments with regard to the statistical models require review by an expert with regard to biostatistics.
Author Response
We thank Reviewer for the time spent in reviewing our manuscript. We also acknowledge the constructive feedback. This particular comment has been taken into consideration.
So, in order to improve the work’s design and to refine the sample of the juvenile group, animals without complete testicular descent were removed, which led to more accurate results as presently all donkeys included in the study (juvenile and mature) showed both testicles in scrotal position.
For this, 8 juvenile donkeys were removed from the sample. This removing implied corrections along the manuscript (tables and text), as the statistics analysis was rebuilt. The changes are now marked in red color. Still, the strength of the Model and the interest of its application did not decrease within the scope of work’s objectives.
Additional revisions:
Considering the interest of the readers on descriptive analysis of the biometric and testicular morphometric data of donkeys, two supplementary Tables (S6 and S7.1. and S7.2.) were added in order to provide this data and to give comparative analysis of juvenile versus mature donkeys.
Reviewer 2 Report
Please see comments for editors
Author Response
No specific comments or suggestions were sent to authors from this reviewer.
Globally, with the major corrections that have been made, the manuscript was improved in its clarity and accuracy.
Finally, we would like to thank to Reviewer for the time spent in reviewing our manuscript.
Reviewer 3 Report
The authors have addressed the questions I raised.
Author Response
We thank Reviewer for the time spent in reviewing our manuscript. We also acknowledge the positive feedback.
Round 2
Reviewer 1 Report
The authors have provided a satisfying revision of their manuscript that is now considered acceptable for publication.
Author Response
Dear Reviewer,
The team responsible for this paper acknowledge the comments from the reviewers and Editor, as they help to improve the quality of our manuscript.
Manuscript was reviewed by a Cambridge University ESOL Examinations Instructor in order to check and correct grammar, typos and impeding mistakes to improve readability.
Likewise, many thanks for Norah Li's editorial work.
We very much thank you.

This manuscript is a resubmission of an earlier submission. The following is a list of the peer review reports and author responses from that submission.
Round 1
Reviewer 1 Report
The current manuscript describes two statistical models for predicting sperm output in correlation with age, weight, testicular morphometry, and some combined biometric indices.
The results are logic and comedies with the previous literature. I have some minor comments:
1- The title:
Use "Semen parameters" instead of "sperm output".
Use "Indices" instead "indexes".
2- There is around 12% differences between the Normal and Motile sperms, what was the source of sperm abnormality? Please provide a supplemental data of Head, Mid-piece, and Tail abnormalities details.
3- In Table 2: correct "Pearson's Correlation Coefficient".
4- Regarding TSN and TMS, they are a function of sperm concentration and volumes, so it will certainly correlate with them. No need to add such big numbers in the analysis.
Author Response
Comments and Suggestions for Authors
The current manuscript describes two statistical models for predicting sperm output in correlation with age, weight, testicular morphometry, and some combined biometric indices.
The results are logic and comedies with the previous literature. I have some minor comments:
RE: we thank the Reviewer 1 for the time spent to review our manuscript and the useful comments. We have addressed the revisions as requested and marked them in blue colour in the manuscript. Below, we provide the point by point answers to the comments of the reviewer. Line numbers refer to the revised version of the manuscript.
1- The title:
Use "Semen parameters" instead of "sperm output".
RE: corrected accordingly in the revised version. The title is now: Bayesian linear regression modelling for sperm quality parameters through age, body weight, testicular morphometry and combined biometric indices in donkeys
Use "Indices" instead "indexes".
RE: we agree and thank this opportune remark. Corrected as suggested in the title and in the text.
2- There is around 12% differences between the Normal and Motile sperms, what was the source of sperm abnormality? Please provide a supplemental data of Head, Mid-piece, and Tail abnormalities details.
RE: In fact, there is 87% of normal sperm morphology, whereas only 72% showed motility. The difference could be attributed to the fact that the motility values presented concern progressive motility (not total motility). A supplementary table S6 provides information on the abnormal sperm morphology details.
3- In Table 2: correct "Pearson's Correlation Coefficient".
RE: thank you, corrected accordingly in table 2 and in the text.
4- Regarding TSN and TMS, they are a function of sperm concentration and volumes, so it will certainly correlate with them. No need to add such big numbers in the analysis.
RE: we appreciate this comment, which has been corrected in the revised version. For instance, the previous Table 2 is now Supplementary Table 3.

Reviewer 2 Report
Accurate determination of the testes volume and prediction of the daily sperm output is valuable information for reproductive management of a stallion irrespective of species or breed. It may be correct that the development of a Bayesian linear regression model for this purpose requires only a small number of data from a statistician’s point of view, but it is certainly of importance that these data are of high accuracy. This is, however, certainly not the case with regard to the data included into the present investigation. The experimental design of the present study (i.e. population of donkeys included into the analysis) suffers from several major problems that question the quality of the data. The manuscript is also rather long and not carefully prepared. It is very difficult to follow. In conclusion, the manuscript does not meet the standards for publication in a renowned scientific journal. It should not get further publication consideration unless the authors correct their data set and revise the manuscript in an appropriate manner.
The following major concerns have to be considered:
- Male donkeys were included into the study irrespective of their reproductive maturity and health. Testis volume and parenchymal weight correlate highly with daily sperm production, therefore testicular volume is a useful predictor of a stallion’s breeding potential. This endpoint, however, is only reliable in postpubertal stallions because in prepubertal males, the histology of the testicular parenchyma is completely different (i.e. no spermatogenesis, different composition of the parenchyma). To the best of my knowledge, there is no information to what degree testicular size in prepubertal stallions is associated to their testicular size and fertility after puberty. In addition, incomplete testicular descent in prepubertal males has to be considered a sign of future reproductive problems and would at least require follow up examinations of the respective animal. Based on this information, the authors have to define clear inclusion and exclusion criteria for the donkeys they use for the Bayesian linear regression model. There is no doubt that at least data from prepubertal donkeys have to be removed from this analysis.
- Information on animals is not very detailed and has to be improved.
- The number of donkeys used for examination of semen characteristics is far too low and heterogenous. Therefore, no appropriate validation of data was performed. There is a very high variation in semen characteristics among ejaculates and among animals (Table 1). It is unclear if there is also a high variation in semen characteristics among ejaculates collected from the same animal. This, however, has to be assumed from the data and is a major bias. Because TSN and TMS are the most important characteristics for predicting fertility, it is of utmost importance that reliable information is included for the model.
- Previous studies have demonstrated (e.g. Love et al. 1991, JRF Suppl. 44:99-105; Pricking et al. 2017, Theriogenology 104:149-155) that the evaluation of testicular size is more challenging than often expected. Therefore, even the assessment of testicular size for the present investigation has to be critically validated. Instead of analysis of correlations between size evaluations made in vivo and after castration, Bland-Altmann analysis should be used.
- The manuscript is not always prepared with the required care and accurateness. Please check your tables carefully (e.g. for dimensions, data for TSN and TMS are certainly given in millions not in billions as stated). Check the references (e.g. page 16, line 29: reference #39 from your reference list is not appropriate in this context!). Numbers have to be correctly stated, i.e. n=40 ejaculates origin from only 8 animals. This is a major difference!
Author Response
Comments and Suggestions for Authors
Accurate determination of the testes volume and prediction of the daily sperm output is valuable information for reproductive management of a stallion irrespective of species or breed. It may be correct that the development of a Bayesian linear regression model for this purpose requires only a small number of data from a statistician’s point of view, but it is certainly of importance that these data are of high accuracy. This is, however, certainly not the case with regard to the data included into the present investigation. The experimental design of the present study (i.e. population of donkeys included into the analysis) suffers from several major problems that question the quality of the data. The manuscript is also rather long and not carefully prepared. It is very difficult to follow. In conclusion, the manuscript does not meet the standards for publication in a renowned scientific journal. It should not get further publication consideration unless the authors correct their data set and revise the manuscript in an appropriate manner.
RE: we thank the Reviewer 2 for the time spent to review our manuscript and the useful comments. We have addressed the revisions as requested and marked them in blue colour in the manuscript. Below, we provide the point by point answers to the comments of the reviewer. Line numbers refer to the revised version of the manuscript. In relation to the remarks in the above comment, we will try to evidence that our data is correct and that the approach is adequate. In relation to the extension and preparation, manuscript was carefully revised.
The following major concerns have to be considered:
- Male donkeys were included into the study irrespective of their reproductive maturity and health. Testis volume and parenchymal weight correlate highly with daily sperm production, therefore testicular volume is a useful predictor of a stallion’s breeding potential. This endpoint, however, is only reliable in postpubertal stallions because in prepubertal males, the histology of the testicular parenchyma is completely different (i.e. no spermatogenesis, different composition of the parenchyma). To the best of my knowledge, there is no information to what degree testicular size in prepubertal stallions is associated to their testicular size and fertility after puberty. In addition, incomplete testicular descent in prepubertal males has to be considered a sign of future reproductive problems and would at least require follow up examinations of the respective animal.
RE: In relation to the prepubertal males, they were all juveniles (less than 14 months), so the fact that some of them did not show complete descent of testicles wasn’t considered pathological, since a recent work point out that between 10 to 14 months of age is still considered the pubertal transition period in donkeys (Rota el al., 2018) and previous works considered that donkeys reach puberty from 10 months to 1.5 years, when testicles descend (Fielding and Krause, 1998). We add this paragraph to discussion (L24 to L28) in order to clarify and now is: “Some of the juvenile donkeys presented incomplete testicular descent, which influenced the dimensions of the gonad, as inguinal testicles were smaller that scrotal. As verified previously [37], donkeys between 10 to 14 months are still in pubertal transition period and, at 19–20 months of age, they are at the end of pubertal transition period, when testis have presumably completed descent”.
- Based on this information, the authors have to define clear inclusion and exclusion criteria for the donkeys they use for the Bayesian linear regression model. There is no doubt that at least data from prepubertal donkeys have to be removed from this analysis.
RE: we thank the opportunity to better explain the tools used in the present study. In order to improve the manuscript and explain the models, substantial and complete information about models has been added to Introduction L73-94 then L99-L116 and also in Model Validity L283--300.
The statistical tools used in this study were chosen to fit the characteristics of the data to be analysed. According to Oravecz and Muth [1], the popularity of Growth curve modelling (GCM) relies in its flexibility in simultaneously analysing both within-individual (e.g., changes with age, change due to intervention, due to natural changes occurring along the life of the individuals, among others) and between-individual effects (i.e., individual differences); in other words, the GCM models inter-individual differences in intra-individual variation. GCM has been successfully used to model the evolution of semen parameters in males from other species, such as boars [2].
An individual-specific growth trajectory, specified as a mathematical function which describes how variables relate to each other over time, captures how an individual uniquely changes.
Growth curve modelling covers situations that range from those for which the function of change is linear to other curvilinear polynomial functions (for instance, quadratic, cubic, etc.) which means modelling does not limit to consider straight-line functional growth.
Beyond handling varying growth functions, GCM can flexibly handle unbalanced designs, meaning study individuals may be measured at different occasions and need not be excluded from analysis, even if some of their measurements are missing.
In these regards, Bayesian inference potentiates the flexibility of GCM given Bayesian analyses do not assume large samples, as it would happen in maximum likelihood estimation (either it is nonparametric or parametric inference), smaller data sets can be evaluated preventing power loss and retaining precision, as suggested by Hox, et al. [3] and Lee and Song [4]. In small sample size conditions, the probability of finding significant results decreases [5]. This limitation often translate into, given power issues, an increased hardness to obtain meaningful results [6]. In this context, Bayesian estimation methods have been reported to require a much smaller ratio of parameters to observations (1:3 instead of 1:5).
- Information on animals is not very detailed and has to be improved.
- R: we agree with the reviewer and thank the suggestion. This part has been corrected as suggested, incorporating a new Table (Table 1) with the data of the donkeys enrolled in the study (L145 of Material and Methods).
- The number of donkeys used for examination of semen characteristics is far too low and heterogeneous. Therefore, no appropriate validation of data was performed. There is a very high variation in semen characteristics among ejaculates and among animals (Table 1). It is unclear if there is also a high variation in semen characteristics among ejaculates collected from the same animal. This, however, has to be assumed from the data and is a major bias. Because TSN and TMS are the most important characteristics for predicting fertility, it is of utmost importance that reliable information is included for the model.
RE: it is inevitable, regrettably, that the male sample is small, due to the breed structure itself. In 2014, the potentially reproductive Miranda male donkey population comprised 72 males (Quaresma et al., 2014), -and nowadays much less, about 40 males-, being the male demography unevenly distributed. This leads to a great variability, which is a reflection of the population demography and should be recognized in the context of such a small, heterogeneous and endangered population like the Miranda donkey breed. But this is not a unique case. For instance, a few examples of the number of male donkeys studied in previous published works on reproductive topics: -Gastal et al. (1997): 6 Nordestina male donkeys; -Miró et al. (2005): 4 Catalonian male donkeys; Contri el al. (2010): 6 Amiata male donkeys; Rota el al. (2012): 2 male Amiata donkeys; Rota el al. (2018): 4 Amiata male donkeys; Ortiz et al. (2015): 6 Andalusian male donkeys. These numbers reflect the problems mentioned above.
In relation to the validation of data, according to Stoltzfus [7], the basic assumptions that must be met for the outputs of regression analyses to be valid, include independence of errors, linearity in continuous variables, absence of multicollinearity, and lack of strongly influential outliers. Additionally, there should be an adequate number of events per independent variable to avoid an overfit model, with commonly recommended minimum “rules of thumb” ranging from 10 to 20 events per covariate. This would be supported by Chen, et al. [8] who suggested the usual minimum number of observations for running a linear regression to be 30 to obtain statistically significant estimates. The same authors would even state that sometimes this requirement cannot be met, for instance when the individuals in the sample are limited, which is common to all donkey breeds [9]. Consequently, the general rule of thumb is that to succeed when conducting a linear regression analysis, the number of observations must not be smaller than 30 or 3x(k+1) where k represents the number of independent variables, hence, the sample size used in the present study fulfils all the assumptions to be used in linear regression analyses.In this context, Bayesian inference maximizes the ability to determine significant effects for relatively limited sample sizes, with these sample limitations reflecting in the broadening of confidence intervals, which must be accompanied by an acceptable Bayes factor value. · Previous studies have demonstrated (e.g. Love et al. 1991, JRF Suppl. 44:99-105; Pricking et al. 2017, Theriogenology 104:149-155) that the evaluation of testicular size is more challenging than often expected. Therefore, even the assessment of testicular size for the present investigation has to be critically validated. Instead of analysis of correlations between size evaluations made in vivo and after castration, Bland-Altmann analysis should be used.
RE: According to Doğan [10], although correlation analyses may erroneously detect the occurrence of incidental relationships, instead of meaningful clinical/biological association, these are the preferable choice in certain contexts. For instance, the same authors state that one of the critical problems in the Bland-Altman analysis -presumably a better option- is the need to meet the assumption of normal distribution. The continuous measurement variables need not to be normally distributed, but their differences should. The data may be tested against the normal distribution using classical methods such as the Shapiro-Wilk test or Kolmogorov-Smirnov test. Additionally, these authors reported the fact that Bland-Altman analysis is not an appropriate method to compare repeated measurements, as those in the present study.
In these regards, Batterham [11] would suggest that in a spreadsheet-based simulation of calibration and validity studies, a Bland-Altman plot of difference vs mean values for the instrument and criterion may show a systematic proportional bias in the instrument's readings, even though none is present. This artifactual bias arises in a Bland-Altman plot of any measures with substantial random error. In contrast, a regression analysis of the criterion vs the instrument shows no bias. The regression analysis also provides complete statistics for recalibrating the instrument, if bias develops or if random error changes since the last calibration. The Bland-Altman analysis of validity should therefore be abandoned in favor of regression, as we did in our study.
- The manuscript is not always prepared with the required care and accurateness. Please check your tables carefully (e.g. for dimensions, data for TSN and TMS are certainly given in millions not in billions as stated).
RE: thank you, corrected accordingly in Table 2.
- Check the references (e.g. page 16, line 29: reference #39 from your reference list is not appropriate in this context!).
RE: references were revised and this reference was corrected, thank you!
- Numbers have to be correctly stated, i.e. n=40 ejaculates origin from only 8 animals. This is a major difference!
R: Donkeys or their testicles were considered the experimental units of the study, while each observational unit comprised each moment that each animals was measured, which were considered the observational units of the study and constituted the study sample. The difference between experimental and observational units must be considered when repeated measurements are taken. The observational unit may be a sample from the experimental unit. For instance, if a single animal is measured several times, each animal is the experimental unit and each time you measure it, is the observational unit (when tabulating the results, each time an animal is measured results in an observation, which is a table row).
As stated in D’Amico, et al. [12], Hintze [13] and Hintze [14], unlike other designs, the repeated measures design has two sample units: between (between experimental units) and within (observational units). In this example, the first (between) experimental unit is a subject. Subject-to-subject variability is used to test the between factor (groups). The second (within) experimental unit is the time period (months). In the above example, the month to month variability within a subject is used to test for the relationship between the testicular morphometry and seminal parameters. The important point to realize is that the repeated measures design has two error components, the between and the within when compared to other types of designs.
In this context, Lorch and Myers [15] suggested an appropriate procedure for analysing the data of repeated measures using a regression design. The data may be organized in an N x J (Subject x Item) matrix in which the entries are the subjects* reading times, and at least four sources of variance should be distinguished: subjects, items (linear), Subject x Linear, and either a single residual term or distinct item (residual) and Subject x.
Residual terms. The critical distinction between this method and others is, that it computes the Subject x Linear term (observation), which is what was perform in our study, hence, each particular observation for each particular animal was considered one element in the sample.

Reviewer 3 Report
Raising awareness of issues affecting rare breeds and identifying novel ways to improve the efficiency of breeding are vital to ensure the survival of individual donkey breeds. As such the data presented in this manuscript could be of value however this will be to a limited audience.
The authors reference that during the selection process, only donkeys with normal sized testicles were included. Can the authors expand on how this was evaluated in view of their comment about the limited information available about testicular biometrics in this breed?
The authors explain why 12 donkeys were part of the study as they were due to be castrated. There is no explanation as to why the other stallions were being examined or for what purposes other than this study sperm collection was taking place.
There is a comment about the difference in size between inguinal and scrotal testicles in the juvenile population but no explanation given for this.
In table 1 no explanation is given for the N values which vary from 217 to 84 - what do these relate to?
Author Response
Comments and Suggestions for Authors
Raising awareness of issues affecting rare breeds and identifying novel ways to improve the efficiency of breeding are vital to ensure the survival of individual donkey breeds. As such the data presented in this manuscript could be of value however this will be to a limited audience.
RE: we thank the Reviewer 3 for the time spent to review our manuscript and the useful comments. We have addressed the revisions as requested and marked them in blue colour in the manuscript. Below, we provide the point by point answers to the comments of the reviewer. Line numbers refer to the revised version of the manuscript.
In relation to this particular comment of the reviewer, we appreciate and agree. In fact, audience could be somewhat limited, but this could not discourage to pursue research in donkeys.
The authors reference that during the selection process, only donkeys with normal sized testicles were included. Can the authors expand on how this was evaluated in view of their comment about the limited information available about testicular biometrics in this breed?
R: in fact, only donkeys with normal sized and symmetrical testicles were included in the study. The perception of normal size was given by our previous experience in this breed (unpublished data) that gave us the perception of normal sized testicles, in its shape and dimensions.
The authors explain why 12 donkeys were part of the study as they were due to be castrated. There is no explanation as to why the other stallions were being examined or for what purposes other than this study sperm collection was taking place.
R: In fact, some of the juvenile donkeys were submitted to castration, usually because of management issues. The other donkeys (juvenile and mature) were examined, measured and collected only with the purpose of getting knowledge of reproductive characteristics of the males as they weren’t previously studied, as stated in Introduction. There were both scientific and pedagogic purposes. The following paragraph has been added:
L126-128 of Material and Methods “Animals have been evaluated with approval and in collaboration with the Association for Study and Protection of the Donkey Breed Burro de Miranda (AEPGA) and in the behalf of a Scientific Protocol of Cooperation signed between both Institutions (Approval 408/Veterinary Teaching Hospital/UTAD)”.
There is a comment about the difference in size between inguinal and scrotal testicles in the juvenile population but no explanation given for this.
R: Thank you, an explanation will be added to Discussion. In fact inguinal testicles are much smaller that inguinal. The new paragraph could be found L24 to L28 of Discussion: “Some of the juvenile donkeys presented incomplete testicular descent, which influenced the dimensions of the gonad, as inguinal testicles were smaller than scrotal. As verified previously [37], donkeys between 10 to 14 months are still in pubertal transition period and, at 19–20 months of age, at the end of pubertal transition period, when presumably testis have completed descent”.
In table 1 no explanation is given for the N values which vary from 217 to 84 - what do these relate to?
R: These are repeated observations, which were the observational units which comprised the study sample. Some information was missing as it was not measured in all the donkeys considered in the study. This was explained in the body text in the previous version of the manuscript “In the present study, comparative observations were taken at different points in time, where the population membership changes over time, but retains some common members, that is, not all the animals which were measured for morphometric parameters were evaluated for semen parameters (partially overlapping samples)”. Furthermore, afterwards, the statistical considerations to process this type of study samples was defined and described.
Lorch and Myers [15] suggested an appropriate procedure for analysing the data of repeated measures using a regression design. The data may be organized in an N x J (Subject x Item) matrix in which the entries are the subjects* reading times, and at least four sources of variance should be distinguished: subjects, items (linear), Subject x Linear, and either a single residual term or distinct item (residual) and Subject x
Residual terms. The critical distinction between this method and others is, that it computes the Subject x Linear term (observation), which is what was perform in our study, hence, each particular observation for each particular animal was considered one element in the sample.

Reviewer 4 Report
Testicular size, because of its easy measurement and high heritability, has become a very sufficient trait to improve male fertility. This manuscript construed Bayesian linear regression modelling for sperm output through age, weight, testicular morphometry and combined biometric indexes in donkeys, which could be helpful for elite jackass selection, while the sample size is a little bit small and age range is to much large, which may decrease the power of their predictive models. In addition, I also have some concerns below:
- The age of donkey analyzed in this study is in a very large range, the author should list how many donkey in each age bracket.
- What is the detail information of 8 mature donkey for ejaculation analysis?
- The author got 24 testis, is density close to one?
- In 2.4.2, only mentioned position analysis, no age.
- In 2.4.4, should be the absolute value of rxy
- Since the author mentioned the donkey have consistent left and right testis at beginning, then they do not need to consider left and right independently.
- In line 246, table 1 cannot be first present after table 3 and table 4
Author Response
REVIEWER 4
Comments and Suggestions for Authors
Testicular size, because of its easy measurement and high heritability, has become a very sufficient trait to improve male fertility. This manuscript construed Bayesian linear regression modelling for sperm output through age, weight, testicular morphometry and combined biometric indexes in donkeys, which could be helpful for elite jackass selection, while the sample size is a little bit small and age range is too much large, which may decrease the power of their predictive models. In addition, I also have some concerns below:
RE: we thank the Reviewer 4 for the time spent to review our manuscript and the useful comments. We have addressed the revisions as requested and marked them in blue colour in the manuscript. Below, we provide the point by point answers to the comments of the reviewer. Line numbers refer to the revised version of the manuscript.
RE: about the sample size, it is inevitable, regrettably, that the male sample is small, due to the breed structure itself. In 2014, the potentially reproductive Miranda male donkey population comprised 72 males (Quaresma et al., 2014), -and nowadays much less, about 40 males-, being the male demography unevenly distributed. This leads to a great variability, which is a reflection of the population demography and should be recognized in the context of such a small, heterogeneous and endangered population like the Miranda donkey breed. But this is not a unique case. For instance, a few examples of the number of male donkeys studied in previous published works on reproductive topics: -Gastal et al. (1997): 6 Nordestina male donkeys; -Miró et al. (2005): 4 Catalonian male donkeys; Contri el al. (2010): 6 Amiata male donkeys; Rota el al. (2012): 2 male Amiata donkeys; Rota el al. (2018): 4 Amiata male donkeys; Ortiz et al. (2015): 6 Andalusian male donkeys. These numbers reflect the problems mentioned above. The age of donkey analysed in this study is in a very large range, the author should list how many donkey in each age bracket.
R: we appreciate this comment. A new table (Table 1) is present in Material and Methods providing information on the donkeys enrolled in the study (L145- Material and Methods)
- What is the detail information of 8 mature donkey for ejaculation analysis?
R: the same commented above, information given in the new table 1.
- The author got 24 testis, is density close to one?
R: according to previous works testis of mammals is close to 1 and our values of testicle weight were close to volume, but only the volumes are shown in Table 1.
-In 2.4.2, only mentioned position analysis, no age.
R: age as removed, thank you (L206- Material and Methods)
- In 2.4.4, should be the absolute value of rxy
R: we agree, indeed when you add two lines, for instance, |0.8| < rxy <|1|, as presented in this study, enclosing each number in mathematics, this means absolute value.
- Since the author mentioned the donkey have consistent left and right testis at beginning, then they do not need to consider left and right independently.
Our results and other authors suggested that testicles behave differently, which suggests they must be considered independently. Examples in the literature corroborate this finding, and these where given in discussion (L78-95) “Mahmoud Ali Omar, et al. [16] reported a similar compensatory effect in the right testicle after the removal of contralateral in donkeys. Other authors have ascribed this compensation to the increase in serum LH and FSH concentrations and, potentially higher intratesticular testosterone [17]. Unilateral orchiectomy has been reported to increase the mean diameter of seminiferous tubules by 21% and of their lumina by 51% [18]. Additionally, two events in line with our results were described. On one hand, a weight compensation was reported for the remaining testis, which has been already described [19]. Secondly, Omar, et al. [20] suggested that the inflammatory response after removal of one testicle may promote the decrease of sperm motility and increase in abnormality percentages. Histological examination of the testis of donkeys after unilateral orchiectomy with scrotum suture revealed hyperplasia of Leydig and Sertoli cells [20]. This had also been reported by Putra and Blackshaw [21], who reported an increase in the number of Sertoli cells and germ cells occupying the seminiferous epithelium after unilateral orchiectomy. Our results suggest that compensation may occur normally without the need of a drastic removal or failure of the testicles, as it was also reported in other species [22]”
- In line 246, table 1 cannot be first present after table 3 and table 4
R: we thank this remark, tables corrected accordingly. Now tables are in the correct sequence.

Reviewer 5 Report
The aim of this study was to develop a Bayesian model for predicting sperm output and other reproductive parameters using testicular biometry in donkeys. This is relevant as pre-requisite for the identification of fertile donkeys for artificial insemination purposes. The authors argued that their choice of a Bayesian model was the robustness of such model with small datasets. However, I have several concerns with this study:
- Development of a prediction equation require some reasonable amount of data to train the model. The dataset used here is so small to provide validity of the model. The choice of a Bayesian approach is not a sufficient replacement for data requirement
- According to the authors, 2 Bayesian models were tested. There is no proper description of the models including mathematical equation in the materials and methods second to ascertain what was done.
- In addition, prediction models need to be validated in an independent population or a seperate dataset from those used for training. This is one way to establish useful of such prediction equation.
- Tonnes of results output from SPSS analysis were presented that has no bearing with the objective of the study and were never discussed. In future, choose only results that are relevant for making conclusion and present such result. Results from pre-analysis of data can be described in the Materials and Method section but not presented as a main finding.
- Poor result presentation. Tables should be presented in the order that they occur and cited accordingly.
- Interpretation of result lacks sufficient evidence and mostly overstated.
- Poor conclusion of findings.
- Throughout the manuscript, grammar is poor and lacks coherence. Sentences were often used out of context.
Author Response
Comments and Suggestions for Authors
The aim of this study was to develop a Bayesian model for predicting sperm output and other reproductive parameters using testicular biometry in donkeys. This is relevant as pre-requisite for the identification of fertile donkeys for artificial insemination purposes. The authors argued that their choice of a Bayesian model was the robustness of such model with small datasets. However, I have several concerns with this study:
RE: we thank the Reviewer 5 for the time spent to review our manuscript and the useful comments. We have addressed the revisions as requested and marked them in blue colour in the manuscript. Below, we provide the point by point answers to the comments of the reviewer. Line numbers refer to the revised version of the manuscript.
- Development of a prediction equation require some reasonable amount of data to train the model. The dataset used here is so small to provide validity of the model. The choice of a Bayesian approach is not a sufficient replacement for data requirement
RE: According to Stoltzfus [7] the basic assumptions that must be met for the outputs of regression analyses to be valid, include independence of errors, linearity in continuous variables, absence of multicollinearity, and lack of strongly influential outliers. Additionally, there should be an adequate number of events per independent variable to avoid an overfit model, with commonly recommended minimum “rules of thumb” ranging from 10 to 20 events per covariate. This would be supported by Chen, et al. [8] who suggested the usual minimum number of observations for running a linear regression to be 30 to obtain statistically significant estimates. The same authors would even state that sometimes this requirement cannot be met, for instance when the individuals in the sample are limited, which is common to all donkey breeds [9]. Consequently, the general rule of thumb is that to succeed when conducting a linear regression analysis, the number of observations must not be smaller than 30 or 3x(k+1) where k represents the number of independent variables, hence, the sample size used in the present study fulfils all the assumptions to be used in linear regression analyses.
In this context, as aforementioned, Bayesian estimation methods have been reported to require a much smaller ratio of parameters to observations (1:3 instead of 1:5) that is Bayesian inference maximizes the ability to determine significant effects for relatively limited sample sizes. With these sample limitations reflecting in the broadening of confidence intervals, which must be accompanied by an acceptable Bayes factor value.
- According to the authors, 2 Bayesian models were tested. There is no proper description of the models including mathematical equation in the materials and methods second to ascertain what was done.
RE: The equations and description for their terms are present in section 3.3.2. Also, the following sentence was added to this section: “The summary of the results for the parameters of validity of both models is reported in Table 8”.
- In addition, prediction models need to be validated in an independent population or a separate dataset from those used for training. This is one way to establish useful of such prediction equation.
RE: The article aimed at comparing models. Thus, statistical validation was performed using the statistical methods that have been recommended by literature. For this reason, we added the following information, also in the manuscript, on the model validity section (2.4.5.1.) (L282-295).
The process of validation and comparison of Bayesian model is fully mathematically described in Geweke [23]. In this context, some authors [24] have suggested a correct proof for model validation should be based on the mean square error (MSE) of the models being evaluated. Additionally, although Mean Square Residual or Error (MSE) have been used ad widely reported to measure how close a regression line is to a set of points, that is how good a certain model fits the data being observed and Minimum Mean-Square Residual or error (MMSE) , mean square prediction error or MSPE (= RSS/no. of observations) was chosen to measure error variation given MSE has been reported to be influenced by the number of parameters [22] in cases of reduced sample sizes [25,26]. Residual sum of squares (RSS) measures the amount of variance in a data set that is not explained by a regression model. If we consider a regression to be a measurement of the strength of the relationship between a dependent variable and an independent variable of a set of independent variables, then the RSS measures the amount of error remaining between the regression function and the data set. A smaller RSS figure represents a regression function. Essentially determines how well a regression model explains or represents the data in the model.
In Bayesian inference, Monte Carlo Standard Error (MCSE) is another measure of accuracy of the chains. It is defined as standard deviation of the chains divided by their effective sample size. MCSE has been reported to be the nonparametric or Bayesian counterpart of MSPE, to be used as the validation criteria in Bayesian Linear Regression model comparison studies [27].
Then afterwards, BIC was calculated as it explains how well the model will predict on new data. Bayesian information criterion (BIC) or Schwarz information criterion (also SIC, SBC, SBIC) was computed as follows,
Where MSPE is the mean squared prediction error, N is the number of observations or records and K is the number of independent parameters of the model.
- Tonnes of results output from SPSS analysis were presented that has no bearing with the objective of the study and were never discussed. In future, choose only results that are relevant for making conclusion and present such result. Results from pre-analysis of data can be described in the Materials and Method section but not presented as a main finding.
RE: we deeply acknowledge this suggestion. For instance, we move the previous Table 2 to the group of Supplementary tables, being now table S3. Also, the text and tables on the normality of data distribution were integrated in Material and Methods Section (L198 to L202) and Supplementary Tables 1 and 2, respectively.
- Poor result presentation. Tables should be presented in the order that they occur and cited accordingly.
RE: we apologize for this inaccuracy. We corrected the order of the tables, which are now in the correct sequence.
- Interpretation of result lacks sufficient evidence and mostly overstated.
RE: thank you for the comment as it gives us an opportunity to improve discussion. We agree that the paragraph on the testicular asymmetry (Discussion section) could generate some uncertainty. Now we provided more clarity to this paragraph, remove sentences that were not so applicable to our work and this fact will help the interpretation of the findings (L78-L96):
- Poor conclusion of findings.
RE: Conclusions have been improved, with inclusion of an additional paragraph that highlights the most important findings of the first part of the work and the conclusions that this work allowed (L124-L129):
“Results of the present work evidences the reliability of ultrasonographic measurements of testis which emphasises its importance and value for obtaining reference values of donkey testicular volumes. Values of testicular volume in Miranda donkey breed suggest similarity to other similar European donkey breeds, as well as sperm output. The gonadosomatic index (GSI) is higher in donkey than in other domestic species previously described, confirming a great reproductive potential of male donkeys”.
The last paragraph (L130-138) paragraph was also improved:
“Results of the present study also suggest that combinations of biometrical and testicular morphometric factors (age, body weight, testicular volume and GSI) will likely improve the predictive accuracy of Models than the factors separately. Based on our findings, it is recommended that, besides biometry data like BW and age, testicular volume and GSI could be systematically taken into consideration and integrated on BSE of donkeys. The present study provides new insights into donkey reproductive biology that may be transferred to ARS strategies. Appropriate use of both Models may be useful to further improve knowledge on the reproductive characteristics of donkey breeds, for clinical purposes and also as a way to direct conservation or selection strategies”
- Throughout the manuscript, grammar is poor and lacks coherence. Sentences were often used out of context.
RE: Manuscript was now revised by a Cambridge ESOL examination instructor to check for typo and grammar mistakes and in order to improve readability.
